# Path entropy-driven design of solid-state electrolytes

Qiye Guan [1] ✉, Kaiyang Wang [2], Jingjie Yeo [2] & Yongqing Cai [1] ✉

The development of high-performance solid-state electrolytes (SSEs) has entered a critical stage, where entropy-driven strategies offer transformative potential for enhancing electrochemical properties. By engineering local environments for conductive ions alongside introducing disorder, these approaches can significantly improve conductivity. However, embracing high-entropy designs does not always guarantee improved performance. Current entropy descriptions oversimplify disorder by accounting solely for host framework configurations, neglecting conductive ion-induced disorder, rendering such descriptions incomplete. Herein, we propose path entropy ($S_p$) as a descriptor that quantifies diffusion pathway diversity, directly capturing diffusional disorder. Combining Markov state model with transition path theory, we reveal the interplay between diffusion pathway diversity of lithium and microscopic local environments in inorganic thiophosphates. Generalizing this path-informative $S_p$ for high-throughput screening, we demonstrate its broad applicability in identifying and designing high-performance SSEs. Our work establishes a critical link between entropy evolution underlying ion conduction and practical entropy-driven design principles.

Recent years have underscored the critical role of high-performance batteries in storing intermittent renewable energy[1]. Driven by the pursuit of all-solid-state metal batteries characterized by high energy density and safety, SSEs with high ionic conductivity have received substantial attention[2,3]. However, the inclusion of many-body and group-concerted motions in ionic conduction results in significantly more complex behavior in SSEs than in single-entity or classical ionic conductors[3–5], rendering dramatic challenges in theoretical and experimental design. In fact, experimentally verified bond breaking and reorganization triggered by ion conduction occur at frequencies up to 40 THz[6], revealing ultrafast fluctuations of thermodynamic variables.

A rigorous thermodynamic formulation understanding ion conduction incorporates the free energy barrier ($\Delta G$), enthalpy change ($\Delta H$), and entropy change ($\Delta S$) at a specific temperature $T$: $\Delta G = \Delta H - T\Delta S$. Consequently, minimization of the $\Delta G$ can be counteracted by an increase in the $\Delta S$, underpinning entropy-driven strategies. By engineering local environments while introducing disorder, ionic conductivity in a specific system can be enhanced through methods such as doping[7,8], vacancy creation[9–11], anion substitution[12–15], etc. While the enthalpies associated with bond formation and breaking are relatively straightforward to quantify, the entropy changes accompanying ionic diffusion remain challenging to characterize. To date, most entropy descriptions in ionic systems focus on structural disorder via configurational entropy, which is often derived from high-entropy alloys[12,16,17]. Alternative descriptors, such as "migration entropy"[18,19], depend solely on lithium-ion hopping/vibrational frequencies. The pathway of diffusive ions has been largely overlooked, leaving the entropy originating from ionic diffusion in SSEs unresolved.

The core challenge lies in quantitatively assessing the entropy arising from all parts in SSEs. Focusing solely on site occupancy information from configurational entropy or metrics such as hopping frequencies[18] and mean square displacement[20] is insufficient. Analysis of diffusion pathways from one site to another, where ions spend a significant fraction of time[21], is fundamental. The diversity of these

[1]Institute of Applied Physics and Materials Engineering, University of Macau, Macau, China. [2]Department of Materials Science and Engineering, Cornell University, Ithaca, NY, USA. ✉e-mail: qiye.guan@connect.um.edu.mo; yongqingcai@um.edu.mo

diffusion pathways is therefore central in quantifying entropy generated by ionic conduction. Inspired by principles of information theory applied to thermodynamics[22,23], we propose that entropy arising from ionic diffusion can be directly quantified through path entropy ($S_p$), a metric that captures information about diffusion pathways traversed by lithium ions. Similar to how configurational entropy encodes structural disorder, path entropy encodes diffusional disorder, which is determined by the routes ions take during their motion. (Fig. 1)

Here, we combine the Markov state model[22] and transition path theory[24,25] to establish a systematic approach for quantifying disorder in SSEs. By decomposing configurational and diffusional entropy contributions, we establish a critical link between local environment engineering and enhanced ionic diffusion. As a proof-of-concept, we employ Li-argyrodite SSEs as model systems. We assess the entropy-driven strategies, such as vacancy creation and anion substitution in Li-argyrodites, through both diffusional and configurational disorder. Critically, conceptions of interpreting the disorder as solely configurational disorder can be misleading, as some strategies (e.g., lithium vacancy creation) introduce minimal configurational disorder but yield substantial improvements in ionic conduction. Taking account of diffusional disorder quantified through path entropy ($S_p$) is therefore crucial as it directly reflects the diffusion capacity of lithium ions. Generalizing this entropy-analysis protocol across a broader range of inorganic sulfide SSEs via high-throughput screening, we identify $Li_4Cr_2C_4SO_{16}$ (with ionic conductivity up to $5.05 \pm 0.23$ mS/cm), which exhibits performance comparable to the well-established argyrodite SSEs.

## Results

### Entropy-driven engineering of argyrodite-type SSEs

Among various SSE families, including halides, oxides, and polymers, sulfides are particularly notable for their high ionic conductivity (σ), typically ranging from $10^{-5}$ to $10^{-2}$ S/cm[14]. Li-argyrodites ($Li_{7-x}BC_{6-x}D_x$, $0 \le x \le 1$, B = P or As; C = S or Se; D = Cl, Br or I), a prominent subtype within the sulfide family with rich polyanionic moieties[26], are distinguished by their facile synthesis process and high σ[27]. In Li-argyrodites, the rigid anion framework $[PS_4]^{3-}$ serves as the structural backbone, accommodating the flexible lithium coordination shells[26,28]. To explore entropy-driven approaches for modulating anion framework−lithium coordination shell interactions, we selected four argyrodite-type SSEs candidates (Fig. 2a−d).

We chose the Li-P-S-Cl system as our model, which includes $Li_5P(S_2Cl)_2$ and $Li_6PS_5Cl$ with $Amm2$ and $F\bar{4}3m$ space groups, designated as LPSCl-I and LPSCl-II (experimentally confirmed superionic phase[29]), respectively. Here, LPSCl-II serves as the benchmark superionic phase, while LPSCl-I acts as a non-superionic reference. To tailor local environments of LPSCl-II via entropy-driven engineering, we implemented two strategies: (i) introducing lithium vacancies and site disorder (sulfur and chloride ions occupying Wyckoff sites 4a and 4c randomly) to form LPSCl-III phase ($Li_{5.5}PS_{4.5}Cl_{1.5}$[9]); and (ii) substituting anion in the framework (replacing P with Si) to generate a new phase as LSPSCl ($Li_{20}Si_3P_3S_{23}Cl$[14,30]). The atomic structure of LPSCl-III was determined using cluster expansion, guided by experimental data[9] (Supplementary Figs. 1−3), while LSPSCl was sourced from Materials Project[30,31].

Diffusion coefficients ($D$) and the σ were calculated (Supplementary Note 1, Supplementary Fig. 4 and 5). Both engineered phases, LPSCl-III and LSPSCl, exhibit enhanced ionic conduction. At 300 K, LPSCl-III and LSPSCl achieve diffusion coefficients of $2.68 \times 10^{-11} \pm 1.52 \times 10^{-13}$ m²/s and $7.22 \times 10^{-12} \pm 1.95 \times 10^{-13}$ m²/s, at least three orders of magnitude higher than pristine LPSCl-II ($4.24 \times 10^{-15} \pm 1.98 \times 10^{-16}$ m²/s) (Supplementary Table 1). Ionic conductivities calculated via the Nernst-Einstein equation align closely with experimental measurements: the σ values for LPSCl-II, LSPSCl,

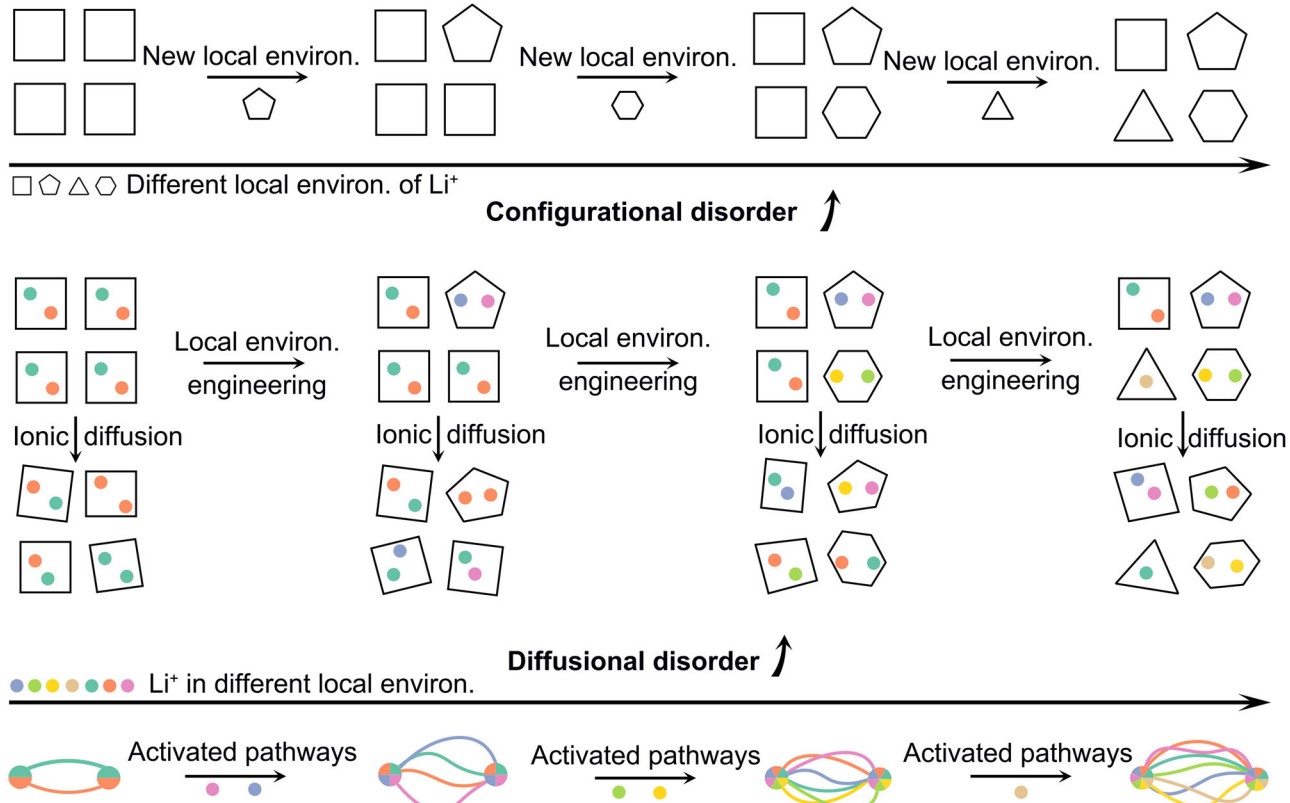

**Fig. 1 | Entropy-driven strategies in solid-state electrolytes.** Schematics of local environment engineering for inducing configurational disorder of the host, and simultaneously diffusional disorder of Li⁺ ions.

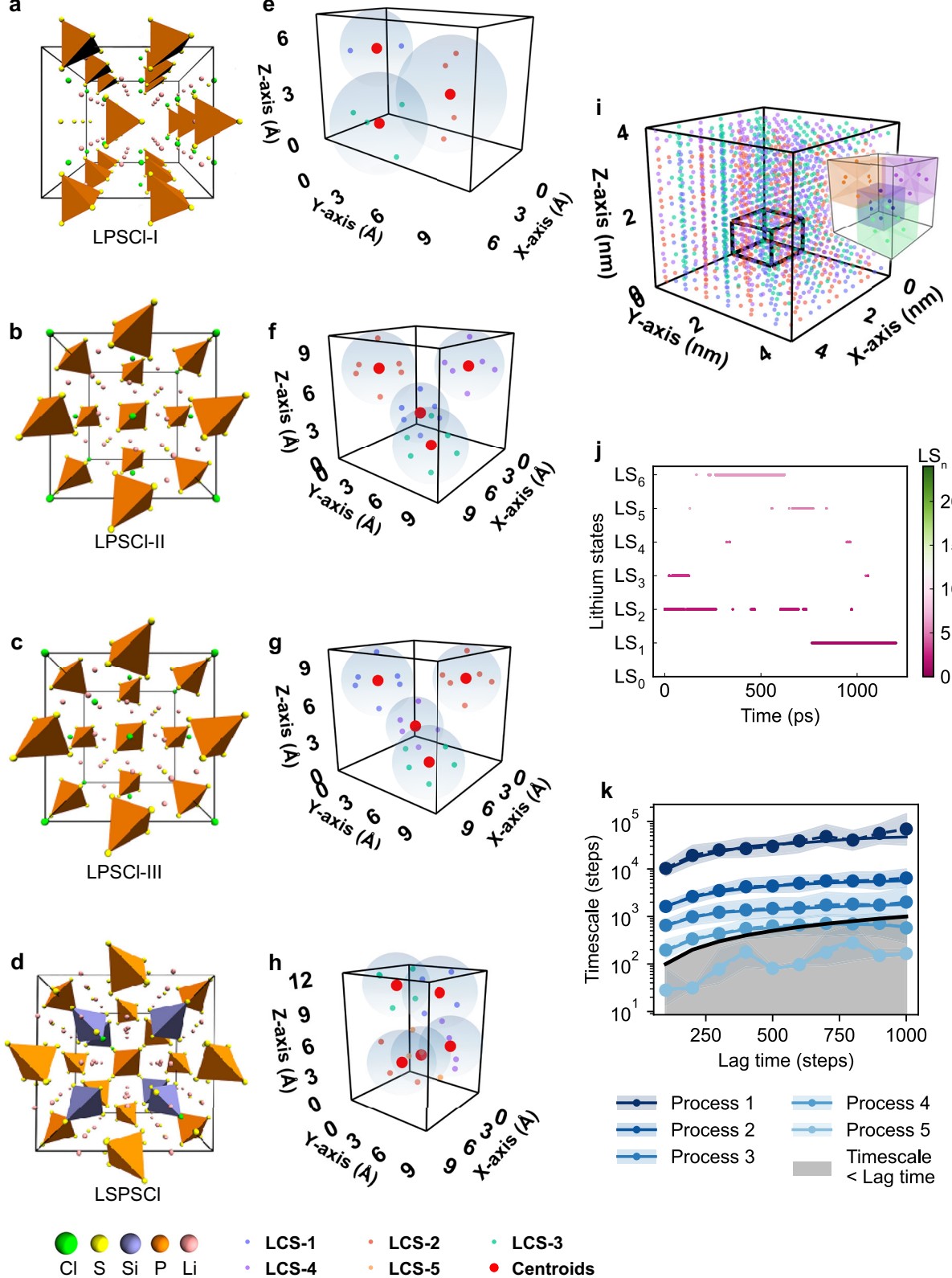

**Fig. 2 | Markov state modeling of lithium-ion dynamics in Li-argyrodites.**
**a**–**d** Crystal structures of $Li_5P(S_2Cl)_2$ (LPSCl-I, *Amm*2 space group), $Li_6PS_5Cl$ (LPSCl-II, $F\bar{4}3m$ space group), $Li_{5.5}PS_{4.5}Cl_{1.5}$ (LPSCl-III), and $Li_{20}Si_3P_3S_{23}Cl$ (LSPSCl), respectively. Orange and blue tetrahedra represent anionic frameworks centered by P and Si, respectively. Representati**e** LCSs in (**e**), LPSCl-I, (**f**), LPSCl-II, (**g**), LPSCl-III, (**h**), LSPSCl. These LCS configurations define different local environments of lithium ions with each occupying a spec**i**fic LS. **i** 3D visualization of lithium-ion distribution in the LPSCl-II supercell employed for simulation. The small cube on the right represents the partitioned LCSs marked by a bold cuboid within the supercell. **j** Decomposing the time evolution trajectory of a single lithium ion over 1.2 ns (20 fs time resolution) into various LSs in LPSCl-II. **k** Implied timescale plot derived from the Markovian process in LPSCl-II. The shaded area around each process represents 95% confidence interval. Region below the solid black line indicates processes occurring faster than the lag time.

and LPSCl-III are $1.53 \times 10^{-3} \pm 1.40 \times 10^{-4}$ mS/cm, (consistent with experimental values[32,33] around $10^{-3}$ mS/cm), $2.37 \pm 0.13$ mS/cm, and $8.42 \pm 0.093$ mS/cm (close to reported value[9] of 9.4 mS/cm), respectively. LPSCl-I manifests the lowest conductivity ($1.10 \times 10^{-7} \pm 3.82 \times 10^{-8}$ mS/cm), confirming its role as a non-superionic reference.

## Modeling lithium-ion diffusion through Markov state models

To capture collective lithium-ion dynamics relevant to battery operation in these engineered SSEs, we employ Markov state models (MSMs)[24,25], an effective way for modeling SSE systems[34,35]. Lithium-ion diffusion in SSEs proceeds via discrete site occupancy/vacancy transitions, forming a continuous-time Markov chain within a discrete state space. To ensure sufficient transitions (a common challenge in modeling Markov chains[25]), we utilize neural network potential-based molecular dynamics (NNMD) simulations trained at the density functional theory (DFT) level (Supplementary Note 2, Supplementary Tables 2-3, Supplementary Figs. 6–11). This enables large-scale molecular dynamics simulations spanning nanometer length scales and nanosecond time scales for our four systems containing thousands of atoms (Supplementary Fig. 12). These trajectories were then used to construct the MSMs.

We first identify discrete lithium-ion spaces via local coordination shells. Ions with proximal positions exhibit similar behavior due to analogous coordination environments, which we define as local coordination shells (LCSs). Following a periodic K-means clustering method (Supplementary Note 3), which determines the optimal number of LCSs, lithium ions are assigned to LCSs characterized by distinct spatial and angular distributions. As shown in Fig. 2e–h, three LCS types are identified in LPSCl-I, four types in LPSCl-II and LPSCl-III, and five types in LSPSCl. We then discretize each LCS into partitioned 3D subspaces representing discrete lithium states (LSs) using classical Voronoi partitioning (Fig. 2i, details in partitioned boundaries and error estimation are provided in Supplementary Note 3). Specifically, a total of 10 LSs are identified for LPSCl-I, 24 LSs for LPSCl-II, 22 LSs for LPSCl-III, and 20 LSs for LSPSCl. For the remaining subspaces, the LS is designated as $LS_0$, representing the possible state lithium ions occupy during diffusion. This methodology enables spatial mapping of lithium-ion trajectories into discrete states (Supplementary Fig. 13), as illustrated in Fig. 2j. After mapping into the lithium state space, we build MSMs for each type of the SSE. As demonstrated in Fig. 2k, we resolve five lithium-ion dynamic processes in the discrete space of LPSCl-II for an individual lithium ion.

A key advantage of the MSM approach is the direct capability to extract kinetic information via mean first passage times (MFPTs)[36], quantifying the average time required for transitions between LSs. The MFPT profiles that capture state transitions across LCSs enable a holistic evaluation for both short-range (intra-LCS) and long-range (inter-LCS) diffusion pathways, revealing how local environments govern lithium-ion diffusion kinetics. While ideal MFPT values require infinite sampling, nanosecond-scale DFT-level simulations effectively capture kinetic differences in ionic diffusion: inter-LCS diffusion is completely blocked in LPSCl-II (Supplementary Figs. 14–17). In contrast, substantial inter-LCS diffusions emerge in LPSCl-III (Supplementary Figs. 18–21) and LPSCl (Supplementary Figs. 22–26), enabling efficient long-range transport. This disparity reveals that tailoring local environments through lithium-vacancy introduction and S/Cl site disorder significantly accelerates inter-LCS transport. For the anion-substituted LSPSCl, while certain states are infrequently visited at room temperature, enhanced LCS diversity also facilitates lithium-ion diffusion across different locations.

## Quantify diffusional disorder through path entropy

Diffusion pathways are strongly correlated with disorder induced by ionic conduction. Through transition path theory (TPT), we characterized lithium-ion flux patterns, including pathway multiplicity and

associated weights, across all LSs. Lithium ions in LPSCl-I are immobile, exhibiting no diffusion pathways across LSs. This result is consistent with its lowest diffusion coefficient among all four systems examined. Local environment engineering in LPSCl-III and LSPSCl yields demonstrably greater pathway diversity compared to pristine LPSCl-II. Specifically, lithium-ion transport from $LS_1$ to $LS_0$ in LPSCl-II follows only several dominant pathways (Fig. 3a). In contrast, LPSCl-III and LSPSCl exhibit expanded pathway networks (Fig. 3b–c), with substantial increases in cross-LCS state transfer events originating from multiple LCSs. This enhanced pathway multiplicity directly correlates with improved ionic conductivity in these two engineered materials.

To quantitatively assess this diffusional disorder, we define path entropy $S_p$ via Shannon entropy formulation[22,23]:

$$S_p = -k_B \sum_{A \neq B} \sum_{i \neq j} \rho_{ij}^{AB} \ln \rho_{ij}^{AB} \tag{1}$$

where $\rho_{ij}^{AB}$ represents transition probability density between two LSs, i.e., $A$ and $B$, quantifying the probabilities of all diffusion pathways through intermediate LSs ($\forall i \in (A \cup B)^c$) (see Methods). The entropic contribution from long-range diffusion, termed escape entropy $S_e$, is defined as transitions originating from the original LCS ($A \in$ LCS) to states outside this LCS ($B \notin$ LCS). As shown in Fig. 3d and Supplementary Tables 4–7, lithium-deficient LPSCl-III exhibits significantly higher $S_p$ ($2598.16 \pm 112.58$ J/mol/K) and $S_e$ ($513.44 \pm 26.86$ J/mol/K) compared to LPSCl-II ($S_p = 415.72 \pm 6.02$ J/mol/K; $S_e = 0$ J/mol/K). This indicates the emergence of enhanced pathway multiplicity, particularly long-range diffusion channels. Introducing additional anions via substitution in LSPSCl also diversifies flux patterns, increasing $S_p$ ($553.91 \pm 54.42$ J/mol/K) and $S_e$ ($91.70 \pm 15.82$ J/mol/K).

Free energy profiles for both inter-LCS and intra-LCS diffusion were constructed to verify a positive correlation between increased diffusional disorder (path entropy) and reduced diffusion barrier by well-tempered metadynamics (WTmetaD). For inter-LCS diffusion, collective variables (CVs) D1 and D2 represent the distances from the selected lithium ion to the centers of their assigned LCSs (Supplementary Note 4.1, Supplementary Fig. 27). LPSCl-III exhibits significantly lower inter-LCS diffusion barriers (~38 kJ/mol) compared to LPSCl-II (~65 kJ/mol) (Fig. 3e–f, Supplementary Figs. 28 and 29). In contrast, intra-LCS motion shows no significant reduction in barrier height in LPSCl-III relative to LPSCl-II (Supplementary Note 4.2, Supplementary Figs. 30–33). This disparity is consistent with enhanced non-stoichiometric flexibility in LPSCl-III, where the increased $S_e$ drives inter-LCS hopping.

## Quantify configurational disorder via configurational entropy

The rapid rotation of the $[PS_4]^{3-}$ moiety in argyrodite SSE systems occurs at ~$10^9$ s$^{-1}$[37], significantly modulating lithium-ion mobility. To compute rotational free energy, we selected the polar angle ($\theta$) and azimuthal angle ($\phi$) of tilted $[PS_4]^{3-}$ tetrahedra as two CVs. (Fig. 4a, Supplementary Note 4.3, Supplementary Table 8). As shown in Fig. 4b and Supplementary Fig. 34, LPSCl-I exhibits restricted rotation and a narrow tilting-state distribution, consistent with its low ionic conduction. In contrast, LPSCl-II (Fig. 4c and Supplementary Fig. 35) and LPSCl-III (Fig. 4d and Supplementary Fig. 36) display unlocked rotation, with LPSCl-III showing significantly lower rotational free energy across the configurational space due to the introduced lithium vacancies. Notably, LSPSCl exhibits a slightly higher rotation barrier of $[PS_4]^{3-}$ tetrahedra than LPSCl-III (Fig. 4e and Supplementary Fig. 37), while a much higher barrier of ~300 kJ/mol for Si-substituted tetrahedra ($[SiS_4]^{4-}$ or $[SiS_3Cl]^{3-}$) (Fig. 4f and Supplementary Figs. 38–40). While elevated barriers restrict bond rotation in $[SiS_4]^{4-}$ and $[SiS_3Cl]^{3-}$, the introduction of Si-substituted tetrahedral species generates distinct configurations of rigid tetrahedra units, which collectively enhance lithium-ion diffusion channel versatility.

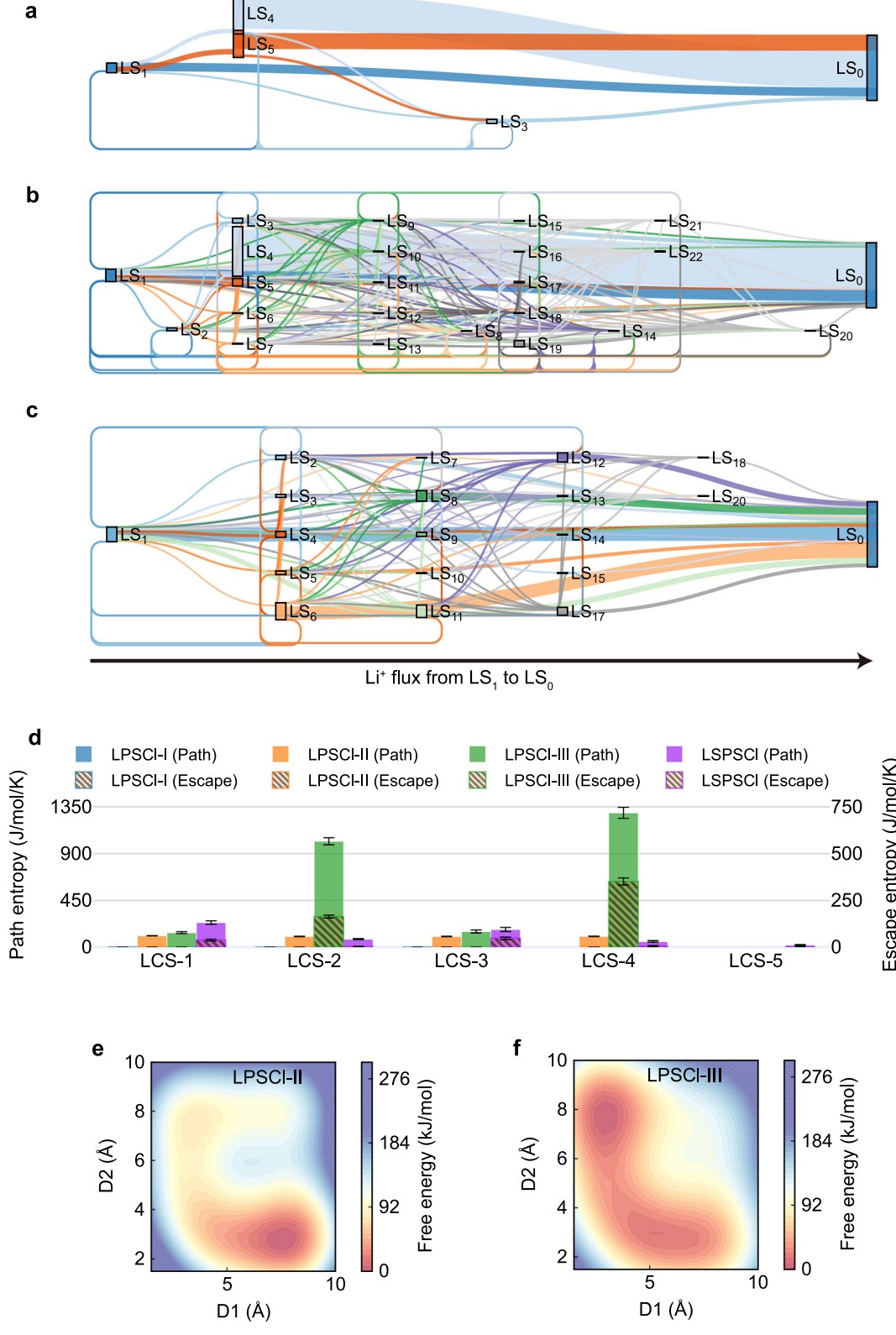

**Fig. 3 | Diffusion pathways of lithium ions.** Decomposed lithium-ion flux from initial state ($LS_1$) to end state ($LS_0$) of (**a**) LPSCl-II, (**b**) LPSCl-III, and (**c**) LSPSCl. **d** Path entropy ($S_p$) and escape entropy ($S_e$) quantifying diffusional disorder in each LCS. Path entropies are presented as mean values over three independent trajectories, each evaluated at three transfer probability cutoffs of 0.14, 0.15, and 0.16. Error bars denote the 95% confidence interval. Free energy profiles for inter-LCS lithium-ion diffusion in (**e**) LPSCl-II and (**f**) LPSCl-III. D1 and D2 denote distances between the selected lithium ion and the centers of two adjacent LCSs, respectively.

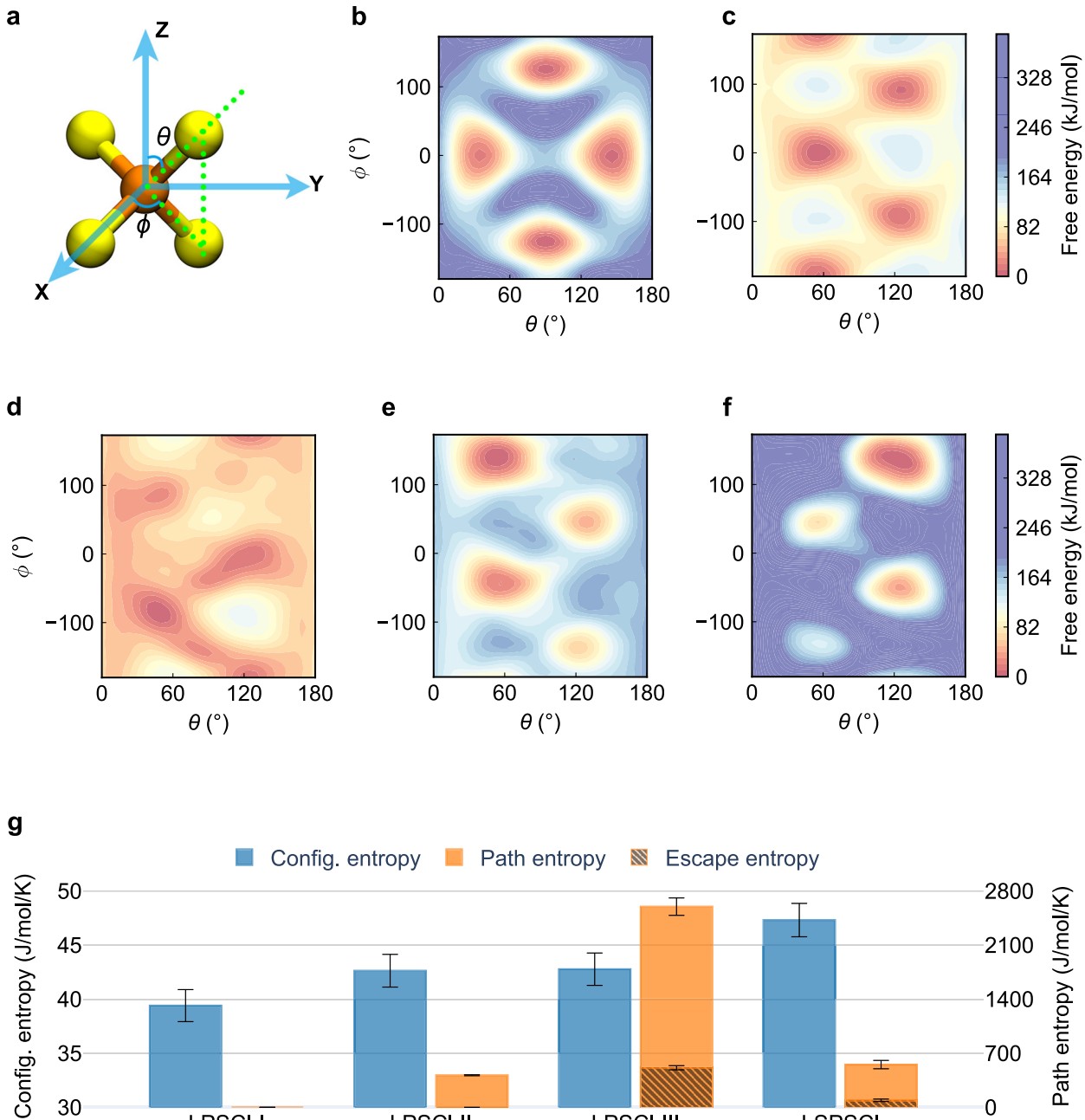

**Fig. 4 | Configurational disorder from anion framework. a** Definition of azimuthal angles $\theta$ (between P-S and Z-axis) and $\phi$ (between projected P-S to XY-plane and X-axis) of tilted $[PS_4]^{3-}$ tetrahedra in cartesian coordinate system. Free energy profiles for tetrahedral rotation in angular space at 300 K of the four argyrodite-type SSE phases: (**b**) LPSCl-I, (**c**) LPSCl-II, (**d**) LPSCl-III, the tilted $[PS_4]^{3-}$ moiety (**e**) and $[SiS_4]^{4-}$ moiety (**f**) in LSPSCl. **g** Comparison of configurational entropy and path entropy for the four phases of argyrodite-type SSEs. Configurational entropies are presented as mean values over three independent trajectories. Path entropies are presented as mean values over three independent trajectories, each evaluated at three transfer probability cutoffs of 0.14, 0.15, and 0.16. Error bars denote the 95% confidence interval.

To quantify anion framework structural disorder, we examined tetrahedral distortion ($\delta_d$), defined as the deviation of $[PS_4]^{3-}$ units from ideal tetrahedral geometry. Configurational entropy $S_c$ is then calculated as[38]:

$$S_c = k_B \ln W(\delta_d) \qquad (2)$$

where $k_B$ is the Boltzmann constant and $W(\delta_d)$ quantifies accessible configurations determined by the $\delta_d$ distribution (see Supplementary Note 5). As shown in Fig. 4g, LPSCl-I exhibits the lowest $S_c$ of 39.41 ± 1.48 J/mol/K with no structural disorder introduced. LPSCl-II

and LPSCl-III have a similar $S_c$ (42.63 ± 1.51 J/mol/K, 42.78 ± 1.50 J/mol/K, respectively), indicating that lithium vacancies or site disorder do not significantly increase structural disorder. Notably, LSPSCl has the highest $S_c$ of 47.31 ± 1.54 J/mol/K, reflecting enhanced disordering due to anion substitution.

**Connection between configurational entropy and path entropy**
Entropy contributions related to ionic conduction in SSEs arise from both the flexible LCSs containing mobile lithium ions and the rigid host framework that forms the structural backbone. Disorder in these systems originates from two components: diffusional disorder associated

with lithium-ion motion and configurational disorder in the host lattice. Consequently, total entropy is partitioned into path entropy $S_p$ from the flexible LCSs and configurational entropy $S_c$ from the rigid host framework. By combining these two components, the system-level disorder can be quantified rather than assessed solely through the disorder induced by the lithium-ion conduction or host framework configurations.

It is noteworthy that while systems with high $S_p$ may exhibit high $S_c$, no direct causal relationship exists between these quantities. Instead, they represent complementary facets of solid-state ionic conductors. Joint analysis of $S_p$ and $S_c$ enables unambiguous identification of entropy gains in entropy-driven-designed systems. Compared to minor variations in the $S_c$ by different entropy-driven strategies (e.g., vacancy introduction), the $S_p$, a metric quantifying diffusional disorder, aligns strongly with ionic diffusion performance. Its values range from 0.0 J/mol/K (LPSCl-I) to 415.72 ± 6.02 J/mol/K (LPSCl-II), 553.91 ± 54.42 J/mol/K (LSPSCl), and 2598.16 ± 112.58 J/mol/K (LPSCl-III) (Fig. 4g). These results confirm that the $S_p$ provides a more direct and quantitatively robust metric for assessing ionic diffusion than the $S_c$. A more nuanced separation of escape entropy $S_e$ from path entropy $S_p$, enables the clear identification of long-range diffusion in LSPSCl (91.70 ± 15.82 J/mol/K) and LPSCl-III (513.44 ± 26.86 J/mol/K).

For Li-argyrodites, configurational disorder is readily quantifiable due to the shared anion framework. However, cross-system calibration remains challenging owing to divergent host frameworks. In contrast, the $S_p$ metric, dependent exclusively on pathway diversity, accounts for a major component of total disorder in ionic systems, establishing it as a universal metric applicable across diverse ionic conductors.

### High-throughput screening of SSE candidates using path entropy

To rigorously validate $S_p$ as a key and general indicator of promising SSE candidates, we performed high-throughput screening using the Alexandria[39] and Materials Project databases[31]. Our initial search encompassed 2524 inorganic sulfides, excluding systems with non-lithium mobile cations (Fig. 5a). Subsequent filtering criteria included: (i) a band gap greater than 2.0 eV ($E_{band} \geq 2.0$ eV) and (ii) an energy above hull ($E_{hull}$) ≤ 85 meV per atom (comparable to LPSCl-II). This reduced the candidate pool to 509 materials. While these candidates satisfied the initial screening criteria for structural and electronic stability, ionic mobility remains a critical factor for SSE performance. To ensure sufficient lithium mobility, we employed NNMD simulations (Supplementary Notes 6.1 and 6.2, Supplementary Fig. 41) to compute mean square displacements (MSDs) of lithium ions. Candidates with measurable lithium-ion mobility, defined as MSDs > 0.010 nm$^2$ over 12.0 ps of NNMD simulation, were retained. This procedure narrowed the list to 27 structurally stable compounds with detectable ionic motion (Supplementary Table 9).

Finally, we implemented a final screening step using $S_p$ and $S_e$ metrics, which quantitatively characterize diffusion pathways diversity and elucidate relative contributions of long-range versus short-range ionic mechanisms. This step identified seven candidate materials (Fig. 5b, Supplementary Fig. 42) with the $S_p > 200.0$ J/mol/K, including experimentally verified SSEs such as argyrodite-type compounds[14,29,30] (LPSCl-II (Li$_6$PS$_5$Cl), Si-substituted LSPSCl[13,14] (Li$_{20}$Si$_3$P$_3$S$_{23}$Cl$_1$), Li$_6$PS$_5$Br, and Li$_7$PS$_6$[40]), thioborate-type[41] Li$_5$B$_7$S$_{13}$[42], Li$_7$P$_3$S$_{11}$[43], rock-salt sulfide Li$_3$NbS$_4$[44], and LGPS-type structures[45] (Li$_{10}$Ge(PS$_6$)$_2$, Li$_{10}$Sn(PS$_6$)$_2$). Notably, we also identified a promising high-performance candidate: Li$_4$Cr$_2$C$_4$SO$_{16}$ ($S_p$ = 263.98 ± 10.42 J/mol/K). This material exhibits a $S_p$ value close to that of Li$_6$PS$_5$Cl (273.52 ± 0.00 J/mol/K), establishing a robust performance baseline.

Fitting the Nernst-Einstein equation (Fig. 5b, Supplementary Fig. 43, Supplementary Note 6.3, Supplementary Table 10), Li$_4$Cr$_2$C$_4$SO$_{16}$ achieves ionic conductivity of 5.05 ± 0.23 mS/cm (Supplementary Fig. 44; activation energy 0.18 eV). Notably, this

conductivity rivals that of LPSCl-III (8.42 ± 0.093 mS/cm), where long-range ionic diffusion is activated via lithium-ion vacancies and site disorder. Li$_4$Cr$_2$C$_4$SO$_{16}$ exhibits 32 lithium sites with sparse distribution compared to LPSCl-II, resulting in 5 connected LCSs (Supplementary Fig. 45). This LCS configuration enables enhanced lithium-ion transport kinetics (Supplementary Figs. 46–50), which fall between the constrained kinetics of LPSCl-II (Supplementary Figs. 14–17) and the fully activated diffusion in LPSCl-III (Supplementary Fig. 18-21). The partially unlocked out-LCS conduction facilitates higher ionic conductivity (Supplementary Fig. 44) despite comparable path entropy to LPSCl-II. The nearly-zero $S_e$ (1.8 J/mol/K; Supplementary Table 9) in Li$_4$Cr$_2$C$_4$SO$_{16}$ confirms predominantly short-range diffusion mechanisms, analogous to pristine LPSCl-II. This characteristic suggests that employing entropy-driven strategies (e.g., vacancy engineering or site disorder) shown in step 6 of Fig. 5a could further significantly enhance its superionic performance.

## Discussion

In summary, by quantifying diffusional disorder through path entropy and configurational disorder through configurational entropy, we elucidate the fundamental principles governing entropy-driven design in SSEs. While configurational disorder is traditionally regarded as the dominant factor in counting system-wide disorder in ionic systems, our analysis reveals that diffusional disorder quantified by path entropy is significantly more pronounced. This distinction is critical: entropy-driven strategies (e.g., lithium vacancy engineering), which induce a minor increment in configurational disorder, can exhibit markedly superior diffusion behavior, outperforming approaches that do increase configurational disorder (e.g., anion substitution). Ignoring diffusional disorder risks mischaracterizing the performance of entropy-driven-designed systems.

We further establish a computationally robust framework for quantifying diffusional disorder, a concept applicable to all inorganic SSEs. Applied to sulfide SSEs, this framework identifies promising candidates such as Li$_4$Cr$_2$C$_4$SO$_{16}$ and highlights their potential for enhanced performance via targeted entropy-driven optimization. By linking the entropy evolution underlying lithium conduction to actionable design principles, this work provides mechanistic insights into superionic behavior and advances entropy-based strategies for next-generation SSE development.

## Methods

### Ab initio molecular dynamics simulations

First-principle calculations have been performed through CP2K software package[46]. The Perdew−Burke−Ernzerhof (PBE) generalized gradient approximation (GGA)[47] with the double-zeta valence polarized basis set and Goedecker−Teter−Hutter pseudopotentials were adopted[48]. The auxiliary plane wave basis set was truncated using a density cutoff of 500 Ry and the van der Waals (vdW) interactions were evaluated through Grimme D3 correction. A time step of 2.0 fs was used in all ab initio molecular dynamics (AIMD) and well-tempered metadynamics (WTmetaD) simulations. The Nosé-Hoover thermostat and Martyna-Tobias-Klein barostat were used with coupling constants of 1.0 ps and 0.5 ps, respectively.

### Identify the crystal structure of LPSCl-III (Li$_{5.5}$PS$_{4.5}$Cl$_{1.5}$)

The crystal structure of LPSCl-III was verified based on the experimental findings[9]. All elements can be classified into Wyckoff sites as follows: Li (48 h), P (4b), Cl (4a and 4c), S (4a, 4c, and 16 e). (see Supplementary Fig. 1) To obtain the exact structure of Li$_{5.5}$PS$_{4.5}$Cl$_{1.5}$, which includes lithium vacancies and site mixture of Cl/S at Wyckoff 4a and 4c sites, we initially employed an enumerative approach to generate possible structures consistent with the site positions. Over a range of cell sizes from 1 to 4, we generated more than 1000 initial structures through enumeration. For 340 structures, we performed

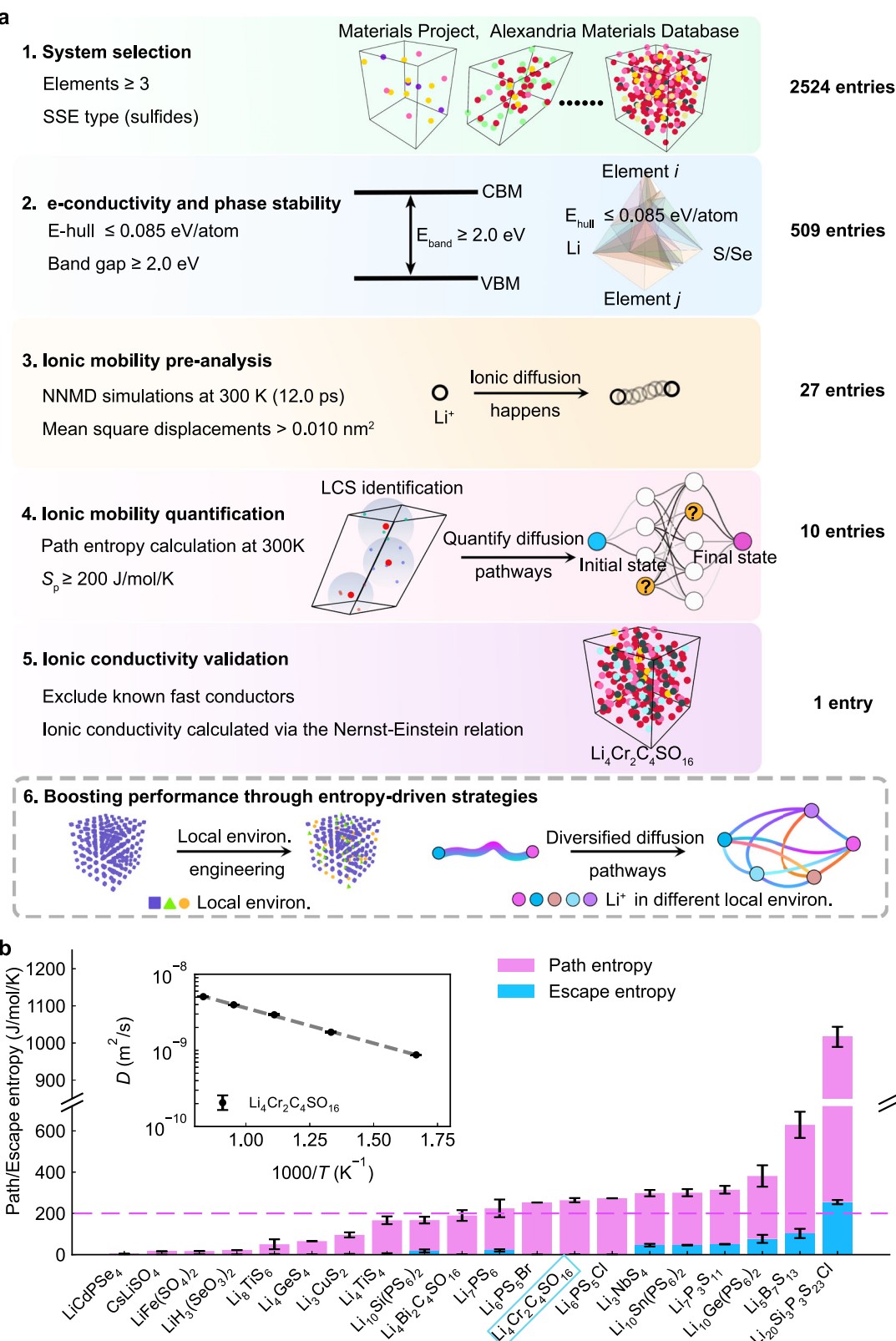

**Fig. 5 | High-throughput screening of potential solid-state electrolytes based on path entropy. a** Flowchart depicting the multistep high-throughput screening procedure for identifying promising SSE candidates. **b** Summary of the path entropy and the escape entropy screening results (top 20 out of 27 candidates) from the step 4 of the workflow. Each path entropy value is the mean over three transfer probability cutoffs of 0.14, 0.15, and 0.16. Error bars denote the 95% confidence interval. Candidates with favorable lithium-ion diffusion characteristics are highlighted with cyan rectangles. The inset shows diffusion coefficient $D$ as a function of the reciprocal temperature for $Li_4Cr_2C_4SO_{16}$. The uncertainty of $D$ was estimated using block averaging with five independent time origins. The reported values are the block-averaged mean, with error bars denoting the 95% confidence interval.

first-principles calculations (Please refer to Supplementary Fig. 2). Next, we employed cluster expansion (CE) to identify the most stable candidates among the target structures (Supplementary Fig. 3). The atomic configuration of the LPSCl-III can be represented by a string of occupation variables, $\{\zeta_1, \zeta_2, \cdots, \zeta_n\}$, where the $\zeta_n$ represents the atomic species occupying the $n^{th}$ site in an $N$-site supercell. The lattice model of the atomic configuration can be written as a sum of cluster interaction functions[49]:

$$H(\zeta) = \sum_{S \subseteq [N]} H_S(\zeta_S) \tag{3}$$

In our CE model, $[N]$ is the set of all site indices, and $\zeta_S$ is the set of all occupation variables for sites in a cluster $S$. The CE model includes pairs of sites separated by less than 6 Å, triples with points less than 5 Å, quadruplets 4 Å apart, and quintuplets 4 Å apart, resulting in a total of 27 correlation functions. Initially, we selected 340 possible candidates to create the CE model, achieving a final mean squared error of 100.05 meV/prim (primitive cell) in predicting the energies. Subsequently, using this model, we predicted around 1100 structures of LPSCl-III as candidates. We then filtered more than ten energetically favorable structures included for neural-network potential training.

## Neural-network potential training

The initial training datasets were prepared through AIMD simulations conducted using the CP2K software package[46]. These simulations were carried out for all four argyrodite-type solid-state electrolytes across temperature ranges from 300 K to 1500 K. Subsequently, neural network-based potentials (NNPs) were trained using the Deep Potential Smooth Edition (DeepPot-SE)[50], which incorporates both angular and radial information from atomic configurations. The detailed active learning for NNP training is outlined as follows (Supplementary Fig. 6): (1) For each type of SSE, we conducted 10.0 ps of AIMD simulations at temperatures of 300 K, 600 K, 900 K, 1200 K, and 1500 K, and a pressure of 1 atm under the NPT ensemble. Subsequently, for each NPT simulation, we selected two frames as candidates for the next MD simulation under the NVT ensemble. Under the NVT ensemble, we employed the On-the-fly Probability Enhanced Sampling (OPES) method[51] with multithermal sampling to enrich the dataset. The internal energy ($U$) of the system was used for defining the CV by $\Delta u$ defined as

$$\Delta u_{\beta'} = (\beta' - \beta)U \tag{4}$$

where $\beta = 1/(k_B T)$ and $k_B$ is Boltzmann constant, $\beta'$ is the inverse thermodynamic temperature to be sampled. For the ten AIMD simulations under the NVT ensemble, we sampled temperature ranges {(0 to 600 K), (300 to 900 K), (600 to 1200 K), (900 to 1500 K)} using multithermal simulations. Each simulation was run for at least 5.0 ps under the NVT ensemble. Note for LPSCl-III, we conducted five MD simulations under the NVT ensemble for all ten candidates predicted to be stable by our CE model. (2) Training the NNP using DeepPot-SE involved extracting energies and forces from the previous MD trajectories. The detailed parameters used for training are provided in Supplementary Table 2. (3) Based on the initial NNP models, a series of NNMD simulations were conducted under both NVT and NPT ensembles using enhanced OPES multithermal-multibaric (MTB). With OPES-MTB, the simulations were biased across a range of temperatures (300 K to 1200 K) and pressures (0.90 atm to 1.10 atm). Candidate configurations were identified based on model deviations of force falling within the range of 0.15 to 0.25 eV/Å. (4) The energies and forces of these candidates were calculated using CP2K, and the resulting data were added to the initial training sets. (5) With the enriched dataset, we iteratively train the new dataset to optimize the model and improve its accuracy. This iterative training process involves updating the neural network potential model based on the new data, refining the model parameters, and repeating the training process until the desired level of accuracy (greater than 99%) is achieved.

To validate the NNPs, we compare the mean absolute errors and radial distribution functions of lithium between the results obtained from AIMD calculations and the predictions made by the NNPs. (Supplementary Table 3, Supplementary Figs. 7–11)

## Markov state model construction

The construction steps of our MSM in studying lithium-ion diffusion are as follows (Supplementary Fig. 13): MD simulations were performed under the NVT ensemble at room temperature (300 K). We then classified LCSs of each SSE structure and discretized them (Supplementary Note 3). Lithium ions in each LCS were uniquely assigned to discretized states. A lithium-ion diffusion trajectory can thus be modeled as a chain of random variables $H_1, H_2, \ldots, H_t, \ldots$ over discrete time moments ($t$) with $H_t$ being randomly one state in the state space $\{X_i\}$. Within the framework of the Markov method, the probability distribution ($p$) of $H_{t+1}$ variable at the time moment $t+1$ is assumed to depend only on the variable at the prior moment (i.e., $t$), thus we have:

$$p(H_{t+1}|H_t, H_{t-1}, \ldots, H_2, H_1) = p(H_{t+1}|H_t) \tag{5}$$

Therefore, the time-series trajectories of lithium atoms are equivalent to a time series of probabilistic variables. The propagation of the probability density function $\mathbf{p}(t)$ over a time interval τ (lag time) is represented as:

$$\mathbf{p}(t + \tau)^T = \mathbf{p}(t)^T \mathbf{T}(\tau) \tag{6}$$

and its explicit form expressed as:

$$\mathbf{p}_j(t + \tau) = \sum_{i=1}^{n} \mathbf{p}_i(t) \mathbb{P}\left(x(t+\tau) \in X_j | x(t) \in X_i\right) \tag{7}$$

where $\mathbf{T}(\tau)$ is the transition probability matrix (TPM) with its component $\mathbf{T}_{ij}$ being the transition probability between $X_i$ and $X_j$ over $\tau$, and its eigenvectors offer insights into the population flux of the dynamic process of lithium site hopping. Consequently, its eigenvalues ($\lambda_i$) denote the timescales of these dynamic processes, which can be depicted in implied time scales (ITS):

$$ITS_i(\tau) = -\frac{\tau}{\ln\lambda_i(\tau)} \tag{8}$$

where $i = 1, 2, 3, 4, \cdots$ represents $i^{th}$ eigenvalue of the $\mathbf{T}(\tau)$. To ensure the reduced state space remains memoryless (or reducing history effect) in capturing the kinetics of lithium hopping, we select a lag time $\tau = 600$ steps (12.0 picosecond). Each LCS in the trajectory contributes to the construction of the MSM, helping to reduce both random and systematic errors in the model. To assess the validity and equality of the lithium hopping MSMs, the Chapman-Kolmogorov (C-K) test was employed. This test evaluates the consistency of transition probabilities between different discrete states over time (Supplementary Fig. 51-53).

## Calculation of lithium flux and path entropy

We implemented TPT to analyze the reactive trajectories of lithium-ion transition. From TPT, we analyzed the flux of the lithium ion that moves from different states[52]. The lithium-ion jumping process was

modeled on the countable state-space $S$ with rate matrix $L = \left(l_{ij}\right)_{i,j \in S}$:

$$\begin{cases} l_{ij} \geq 0 & \forall i, j \in S, i \neq j \\ \sum_{j \in S} l_{ij} = 0 & \forall i \in S \end{cases} \qquad (9)$$

where $l_{ij}$ represents the process jump from state $i$ to state $j$. Given the initial state of lithium at $i$ and the final migrated state at $j$, TPT can compute the reactive flux between any two nonempty, disjoint subsets (e.g., A and B) of the state-space $S$. The reactive trajectory $P$, a set including all the ordered sequences $P_n$ generated from the $n^{th}$ transition between initial state A and end state B, is defined as:

$$P = \bigcup_{n \in \mathbb{Z}} P_n \qquad (10)$$

Then for this reactive trajectory $P$, the discrete forward committor $q^+ = (q_i^+)_{i \in S}$ is defined as the probability that the process starting in $i \in S$ will first evolve toward B rather than A. Similarly, the probability of the process arriving in state $i$ last came from A rather than B is the backward committor $q^- = (q_i^-)_{i \in S}$. We can calculate the probability current of reactive trajectories $P$[25]:

$$f_{ij}^{AB} = \begin{cases} \pi_i q_i^- l_{ij} q_j^+, & \text{if } i \neq j \\ 0, & \text{otherwise} \end{cases} \qquad (11)$$

where $\pi_i$ is the unique stationary distribution. The flux between different states is conserved through:

$$\sum_{j \in S} (f_{ij}^{AB} - f_{ji}^{AB}) = 0 \quad \forall i \in (A \cup B)^c \qquad (12)$$

After having transition flux matrix $f_{ij}^{AB}$, the corresponding probability density $\rho_{ij}^{AB}$ is calculated through L1 normalization for all the components. Ionic transfer between different states $i$ and $j$ is considered effective if the probability exceeds 0.15. The standard error of path entropy is calculated within the transfer probability range of [0.14, 0.16]. For four types of SSE, three independent simulations were performed to compute probability densities of the net transition flux and associated path entropies. The mean path entropies fall within a 99.5% confidence interval.

### Free energy calculation through well-tempered metadynamics

We utilize well-tempered metadynamics (WTmetaD) simulations to investigate lithium-ion diffusion. Various CVs are designed to capture migration behaviors in distinct local environments. For specifics on CV design (Supplementary Figs. 27 and 30) and simulation parameters (Supplementary Table 8), please refer to Supplementary Note 4.

## Data availability

The NNMD trajectories and NNPs of the LPSCl-I, LPSCl-II, LPSCl-III, and LSPSCl are available at https://doi.org/10.5281/zenodo.18829656. Source data are provided as a Source Data file. Source data are provided with this paper.

## Code availability

The sample code to perform the analysis is available at GitHub (https://github.com/DXiming/entropy-driven-SSE) and Zenodo (https://doi.org/10.5281/zenodo.18851246)[53].

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

## Acknowledgements

We thank C. Oses for insightful comments and suggestions on an early version of this manuscript. Q.G. thanks M. Lei for helpful discussions on figure design. This work was supported by the National Natural Science Foundation of China (Grant 22022309), and Natural Science Foundation of Guangdong Province, China (2024A1515011161), University of Macau (MYRG-GRG2024-00028-IAPME, MYRG-GRG2025-00045-IAPME), and Science and Technology Development Fund from Macau SAR (0122/2024/AFJ, 0120/2023/RIA2). This work was performed in part at the High-Performance Computing Cluster (HPCC), supported by the Information and Communication Technology Office (ICTO) of the University of Macau.

## Author contributions

Q.G. planned the project with Y.C.; Q.G. designed and performed calculations. This paper was written by Q.G. and revised by Y.C., J.Y., and K.W.

## Competing interests

The authors declare no competing interests.
