## [Transparent Peer Review file · Nature Communications]

Path Entropy-driven Design of Solid-State Electrolytes

Corresponding Author: Professor Yongqing Cai

Version 0:

Reviewer comments:

Reviewer #1

(Remarks to the Author)

Dear authors,

Thank you for the insightful manuscript. It is very well written and indeed the newly defined path entropy is a central parameter to decouple merely the configurational entropy from potential enhancement in ionic diffusion in compositionally complex or high entropy materials.

The manuscript can be accepted for publication after addressing the following questions:

1. Line 79: $\text{Li}_4\text{Cr}_2\text{C}_4\text{SO}_{16}$ is a computationally identified material. What is the main limitation for its synthesis and experimental validation? Please clarify.
2. Line 149: Are these total LSs or per LCS?
3. Line 155: How many dynamic processes were identified for the other compositions? Is this information included somewhere else?
4. Line 233: Please elaborate on the distinct rotation behaviour, if a higher barrier was calculated. Other factors previously calculated like the inter-LCS diffusion seem to be more relevant for enhancing ion conduction in LSPSCI than tetrahedra rotation? Please discuss in more detail.
5. Why do S_p and S_e in the high-throughput screening for LSPSCI differ from previous calculations shown in Fig.4?
6. Line 461: Why is the standard error in the same order as the probability? Or is an error of 0.01 acceptable in this case? Please clarify.
7. How does the model consider the influence of the electric field? Experimentally, long range ionic conductivity is mostly determined by applying a bias (AC). Is it, nevertheless, possible to correlate the simulated and experimental findings? Please clarify in a short discussion.

(Remarks on code availability)

Reviewer #2

(Remarks to the Author)

In this paper, the authors proposed a descriptor quantifying diffusion pathway diversity—path entropy (S_p), addressing the limitation of traditional configurational entropy that focuses on structural disorder. Through combining Markov state models (MSMs) with transition path theory (TPT), this work revealed the correlation between local environments and ionic diffusion in solid-state electrolytes. Furthermore, this work developed a path entropy-based high-throughput screening process for inorganic thiophosphates, enabling efficient identification of high-performance candidate materials.

In conclusion, I recommend that this work can be considered to publish in Nature Communications after Major revision.

Several comments and questions are included as follow:

1. Why did this work focus merely on verifying the strategies of lithium vacancy creation and anion substitution, instead of exploring other entropy-driven approaches (e.g., cation doping or multicomponent integration)? By the way, investigating these additional methods could further demonstrate the general applicability of pathway entropy as a design criterion.

2. The authors defined escape entropy (Se) as “the contribution from transitions from the original LCS to external states”, but failed to clarify the standard for defining the “original LCS” (i.e., how to define the boundaries of LCS). We suppose that relevant numerical standards can be added.
3. Based on the last review, the gap of LCS among LPSCI-I, II, III, and LSPSCI (3/4/4/5 types) and whether it is related to the intrinsic structure of the materials are expected to be explained.
4. In the flowchart that depicting the multistep high-throughput screening for identifying promising SSE candidates (Fig.5a), the way to determine the threshold setting of “MSD > 0.018 nm²” in Step 3 remains vague. Similarly, in Step 4, why “200 J/mol/K” was chosen as the critical value? Furthermore, is this value universal or specific for sulfide SSE systems?
5. It is necessary for authors to give an in-depth analysis of the fundamental relationship among Sp, Se and Sc. For example, several systems (e.g., LSPSCI) have both high configurational entropy and high path entropy, while others (e.g., LPSCI-III) show low configurational entropy but extremely high path entropy. Is this discrepancy related to the changes in lithium coordination environment?
6. For the novel candidate material Li₄Cr₂C₄SO₁₆, only ionic conductivity at 300 K is provided. To determine its activation energy and high-temperature stability, the temperature dependant EIS (e.g., Arrhenius plot) is suggested to supplement.
7. please confirm whether the error value exceeds the average within the value “8.42 ± 9.34×10⁻² mS/cm”. If it is wrong, please correct it. Otherwise, please explain the reason for the excessive error.

(Remarks on code availability)

Reviewer #3

(Remarks to the Author)

In this manuscript, the authors introduce SP as a novel descriptor to quantify the diffusional disorder of lithium ion pathways in SSEs, using a combination of the MSM, TPT and NNMD. The goal is to establish SP as a universal design principle for HTS of high-performance SSEs. The integration of information theory to analyze the complexity of ion diffusion pathways is a potentially transformative approach, offering a thermodynamic perspective on transport dynamics. Here are some comments:

1. The electrochemical stability is the primary hurdle for practical SSEs. The work solely focuses on transport dynamics. Any proposed material must be stable under synthesis and operating conditions. The manuscript completely lacks analysis of stability of the predicted phases.
2. The manuscript identifies Li₄Cr₂C₄SO₁₆ as a high-performance candidate, yet its calculated diffusion disorder is reported as low, similar to the poorly conducting reference phase. This contradiction must be resolved through deep microstructural and mechanistic analysis.
3. The accuracy of MSM is sensitive to input parameters. The manuscript must prove the statistical robustness of its underlying simulations.
4. Provide a more in-depth microphysical explanation for the complementarity between structural configurational entropy and pathway entropy

(Remarks on code availability)

Version 1:

Reviewer comments:

Reviewer #1

(Remarks to the Author)

Dear Authors,

thank you for thoroughly addressing all comments and making all necessary changes to the manuscript and supporting information.

I suggest accepting the manuscript for publication.

(Remarks on code availability)

Reviewer #2

(Remarks to the Author)

The authors have addressed the comments well. I recommend that this manuscript could be published in Nature Communications.

(Remarks on code availability)

Reviewer #3

(Remarks to the Author)

(Remarks on code availability)

The authors have addressed several of my previous concerns. However, before the manuscript can be recommended for publication, there are two critical technical points regarding the simulation settings and data reliability that must be clarified:

1. In the SI, the authors stated that the energy cutoff for the projector augmented wave method was set to 450 eV. However, for systems containing Li, the standard PAW potentials typically recommend a maximum cutoff of approximately 499.034 eV for high-precision calculations. Could the authors justify the use of 450 eV, which is lower than the recommended value for Li? Please provide evidence to demonstrate that this choice does not compromise the accuracy.
2. On lines 302-303, the author stated "Candidates with measurable lithium-ion mobility, defined as MSDs > 0.018 nm² over 6.0 ps of NNMD simulation". A 6.0 ps window is relatively short for extracting diffusion properties in complex solid-state electrolytes. More importantly, the authors must ensure that the system has reached both thermodynamic and kinetic equilibrium before the commencement of the MSD data collection.

Version 2:

Reviewer comments:

Reviewer #3

(Remarks to the Author)

The revised methodology and accompanying data are now presented with sufficient rigor to fully substantiate the paper's conclusions. In light of these improvements, I recommend the manuscript for publication.

(Remarks on code availability)

Response to Reviewers

Reviewer #1 (Remarks to the Author):

Dear authors,

Thank you for the insightful manuscript. It is very well written and indeed the newly defined path entropy is a central parameter to decouple merely the configurational entropy from potential enhancement in ionic diffusion in compositionally complex or high entropy materials.

The manuscript can be accepted for publication after addressing the following questions:

Re: We sincerely appreciate your thoughtful appreciation of our work. We believe our approach using path entropy to design new solid-state electrolytes will be valuable for the broader scientific community. Below, we have carefully addressed all questions you have raised, with original manuscript sentences underlined and **revised text or new results highlighted in red**.

1. Line 79: *Li₄Cr₂C₄SO₁₆ is a computationally identified material. What is the main limitation for its synthesis and experimental validation? Please clarify.*

Re: Thank you for your specific inquiry regarding the synthesis limitations of Li₄Cr₂C₄SO₁₆. The energy above hull (0.070 eV/atom) of Li₄Cr₂C₄SO₁₆ is lower than Li₆PS₅Cl (0.083 eV/atom), indicating better thermodynamic stability compared to argyrodite systems. Regarding synthesis limitations, the decomposition products were predicted to be LiCr(CO₃)₂ and Li₂SO₄ (Materials Project; [APL materials 1.1 (2013)]). Hence, potential synthesis strategies could involve these precursors. Notably, while Li₂SO₄ has been experimentally synthesized, LiCr(CO₃)₂ (mp-770839) has not been validated according to the Materials Project database. However, similar structures like dolomite (CaMg(CO₃)₂) has already been experimentally verified [Sedimentology 62, 1749 (2015)], which could inform precursor synthesis for LiCr(CO₃)₂.

2. Line 149: *Are these total LSs or per LCS?*

Re: These are total LSs, not per LCS. To clarify this, we have revised the sentence in the revised manuscript to:

“Specifically, a total of 10 LSs are identified for LPSCI-I, 24 LSs for LPSCI-II, 22 LSs for LPSCI-III, and 20 LSs for LSPSCI.” (line 151)

3. Line 155: *How many dynamic processes were identified for the other compositions? Is this information included somewhere else?*

Re: Thank you for your thoughtful inquiry. We previously omitted similar plots for LPSCI-III and LSPSCI due to space constraints. In the revised manuscript, we have now added back the implied timescale and C-K test plots for these systems. As shown in

Supplementary Fig. 52 and **53**, 10 dynamic processes were identified for one selected lithium ion in LPSCI-III and 4 dynamic processes for one selected lithium ion in LSPSCI. We have also added the following cross-reference in the revised manuscript:

“This test evaluates the consistency of transition probabilities between different discrete states over time (**Supplementary Fig. 51-53**).” (line 458)

Supplementary Figure 52. Chapman-Kolmogorov tests for one lithium ion in LPSCI-III.
a, Time evolution trajectory of a single lithium ion over 1.0 ns (20 fs time resolution)

through various lithium states within LPSCI-III. **b**, Implied time scale plot of Markov process in LPSCI-III. **c**, Chapman-Kolmogorov test for the 11 most populated microstates.

Supplementary Figure 53. Chapman-Kolmogorov tests for one lithium ion in LSPSCI. a, Time evolution trajectory of a single lithium ion over 1.5 ns (20 fs time resolution) through various lithium states within LSPSCI. b, Implied time scale plot of Markov process in LSPSCI. c, Chapman-Kolmogorov test for the 5 most populated microstates.

4. Line 233: Please elaborate on the distinct rotation behaviour, if a higher barrier was calculated. Other factors previously calculated like the inter-LCS diffusion seem to be more relevant for enhancing ion conduction in LSPSCI than tetrahedra rotation? Please discuss in more detail.

Re: Thank you for your question regarding the difference between rotation behavior and inter-LCS diffusion.

(1) We have calculated the rotation barriers for all types of tetrahedral units in the LSPSCI system. These units fall into three distinct types:

- a. The pristine $[\text{PS}_4]^{3-}$ unit (shared across all LPSCI systems), with free energy profiles plotted in **Fig. 4e**.
- b. The Si-substituted $[\text{SiS}_4]^{4-}$ and $[\text{SiS}_3\text{Cl}]^{3-}$ (exclusive to LSPSCI), with free energy surfaces shown in **Fig. 4f** and **Supplementary Fig. 40**, respectively.

We have cross-referenced these results in the revised manuscript:

“Notably, LSPSCI exhibits a slightly higher rotation barrier of pristine $[\text{PS}_4]^{3-}$ tetrahedra than LPSCI-III (Fig. 4e and Supplementary Fig. 37), while a much higher barrier of ~ 300 kJ/mol for Si-substituted tetrahedra ($[\text{SiS}_4]^{4-}$ or $[\text{SiS}_3\text{Cl}]^{3-}$) (Fig. 4f and Supplementary Fig. 38-40).” (Lines 230-233)

Fig. 4. Configurational disorder from anion framework. Free energy profiles for tetrahedral rotation in angular space at 300 K of the four argyrodite-type SSE phases: (d) LPSCI-III, the tilted $[\text{PS}_4]^{3-}$ moiety (e) and $[\text{SiS}_4]^{4-}$ moiety (f) in LSPSCI.

Supplementary Figure 40. Rotational free energy profile of the $[\text{Si}_3\text{Cl}]^{3-}$ units in LSPSCI.

Comparing these free energy profiles, which exhibit distinct global minima at different polar and azimuthal angles, we conclude that the introduction of Si-centered tetrahedral units raises unique rotation behavior directly linked to configurational disorder. In the revised manuscript, we have added the following clarification:

“Notably, LSPSCI exhibits a slightly higher rotation barrier of $[\text{PS}_4]^{3-}$ tetrahedra than LPSCI-III (Fig. 4e and Supplementary Fig. 37), while a much higher barrier of ~ 300 kJ/mol for Si-substituted tetrahedra ($[\text{SiS}_4]^{4-}$ or $[\text{SiS}_3\text{Cl}]^{3-}$) (Fig. 4f and Supplementary Fig. 38-40). While elevated barriers restrict bond rotation in $[\text{SiS}_4]^{4-}$ and $[\text{SiS}_3\text{Cl}]^{3-}$, the introduction of Si-substituted tetrahedral species generates distinct configurations of rigid tetrahedra units, which collectively enhance lithium-ion diffusion channel versatility.” (Lines 230-236)

(2) Regarding the second question (“Other factors previously calculated like the inter-LCS diffusion seem to be more relevant for enhancing ion conduction in LSPSCI than tetrahedra rotation?”):

Yes, inter-LCS or intra-LCS diffusion would generally be more relevant for ion conduction in SSEs (not just LSPSCI) than tetrahedral rotation. As discussed in the section “Quantify diffusional disorder through path entropy”, we focused on LPSCI-II and LPSCI-III because they share a similar LCS, enabling direct comparison. For LSPSCI, however, the number and type of LCS are fundamentally distinct from LPSCI-II, making the construction of analogous diffusion collective variables infeasible. Crucially, the primary purpose of calculating rotation barriers here is to provide readers with an intuitive foundation for understanding how configurational entropy is built.

5. Why do Sp and Se in the high-throughput screening for LSPSCI differ from previous calculations shown in Fig.4?

Re: Thank you for your thoughtful question regarding this value difference. This distinction arises from different neural-network potential (NNP) (MACE, [Adv. Neural Inf. Process. Syst. 35, 11423-11436 (2022)]) used in the high-throughput screening section.

For the high-throughput screening, we use a universal NNP, MACE-MPa potential (a PBE+U DFT-level foundation model; **Supplementary Note 6.1**), which is applicable across nearly all materials. In contrast, for the other analyses (specific to our four argyrodite SSEs), we trained DeepMD-kit NNP using high-quality DFT data and active learning (Method section: **Neural-network potential training**).

To validate the robustness of MACE-MPa NNP for high-throughput screening, we have:

- a. compared radial distribution functions (RDFs) with our datasets (**Supplementary Fig. 41**), confirming consistency;
- b. verified that path entropy trends remain similar across both potentials in argyrodite systems (**Supplementary Fig. 42**).

Critically, for rational comparison in the high-throughput screening section, we explicitly reproduced LSPSCI (mp-1040451) and LPSCI-II (mp-985592) under the same MACE NNP, ensuring the analysis remains methodologically sound and directly comparable.

Here are the details in Supplementary data explaining the MACE NNP we have used:

“6.1. Validation of MACE neural-network potential.

Neural network potential-based molecular dynamics simulations were performed utilizing the MACE pretrained foundational model (medium size)³. The accuracy of this model was assessed by comparing radial distribution function (RDF) calculations to analogous results from AIMD, as shown in (Supplementary Figure 41). While the MACE foundational model’s accuracy is somewhat lower than that achieved with actively trained DeepMD-kit models (Supplementary Figure 11), it effectively captures the characteristic RDF peaks and demonstrates broad applicability across a wide range of materials, which is essential for high-throughput screening. Furthermore, consistent results for path entropy were obtained using both MACE and DeepMD-kit (Supplementary Figure 42), providing strong validation for the suitability of the MACE foundational model in this high-throughput context.”

Supplementary Figure 41. Radial distribution function plot of lithium with P, S, Li, and Cl elements at 300 K under NVT ensemble from MACE potential³ for **a**, LPSCI-II, **b**, LSPSCI, and **c**, LPSCI-III. The dotted and solid lines represent the calculation from DFT and MACE potential, respectively.

Supplementary Figure 42. Path entropy comparison of NNMD results from DeepMD and MACE potentials.

6. Line 461: Why is the standard error in the same order as the probability? Or is an error of 0.01 acceptable in this case? Please clarify.

Re: Thank you for your thoughtful question and for catching this important clarification point. The range [0.14, 0.16] refers to the probability values (not standard errors) for which ionic transfer is considered effective. Specifically, we define effective ionic transfer as occurring when the probability exceeds 0.15. We have updated the manuscript accordingly to ensure no ambiguity remains:

“Ionic transfer between different states i and j is considered effective if the probability exceeds 0.15. The standard error of path entropy is calculated within the transfer probability range of [0.14, 0.16].” (Lines 482-484)

7. How does the model consider the influence of the electric field? Experimentally, long range ionic conductivity is mostly determined by applying a bias (AC). Is it, nevertheless, possible to correlate the simulated and experimental findings? Please clarify in a short discussion.

Re: Thank you for your insightful question. We appreciate this opportunity to clarify the scope of our model. The assumption that most periodic units remain stable is indeed foundational to our approach; this is consistent with the framework used in our metadynamics simulations, where external bias potentials were applied to lithium atoms to explore free energy surfaces.

In experimental context, the AC electric field (e.g., 50 Hz) reverses direction 100 times per second (50 cycles per second). This time-dependent behavior can be directly implemented in simulations using tools like “**fix efield**” command in LAMMPS, which adds time-varying external potentials. Crucially, by integrating neural network potentials, we can also achieve realistic simulation of thousands of atoms within seconds. This aligns with the need for efficient exploration of electric field effects in complex systems.

We believe this approach is highly applicable to real-world scenarios where periodic stability dominates, and we are eager to apply it in future work to bridge computational findings with experimental observations.

We would like to thank Reviewer #1 again for the useful suggestions which are important for the improvement of our work.

Reviewer #2 (Remarks to the Author):

In this paper, the authors proposed a descriptor quantifying diffusion pathway diversity—path entropy (S_p), addressing the limitation of traditional configurational entropy that focuses on structural disorder. Through combining Markov state models (MSMs) with transition path theory (TPT), this work revealed the correlation between local environments and ionic diffusion in solid-state electrolytes. Furthermore, this work developed a path entropy-based high-throughput screening process for inorganic thiophosphates, enabling efficient identification of high-performance candidate materials. In conclusion, I recommend that this work can be considered to publish in Nature Communications after Major revision. Several comments and questions are included as follow:

Re: We sincerely thank Reviewer #2 for his/her thorough review of our work and for highlighting how our approach efficiently identifies high-performance candidates. We appreciate her/his recommendation for acceptance after major revisions that could significantly improve the generality and robustness of our calculations. Below, we have carefully addressed all questions Reviewer #2 has raised, with original manuscript sentences underlined and **revised text or new results highlighted in red**.

1. Why did this work focus merely on verifying the strategies of lithium vacancy creation and anion substitution, instead of exploring other entropy-driven approaches (e.g., cation doping or multicomponent integration)? By the way, investigating these additional methods could further demonstrate the general applicability of pathway entropy as a design criterion.

Re: We sincerely appreciate your insightful comments regarding exploring other entropy-driven strategies. We fully agree that incorporating additional entropy-driven methodologies would further strengthen the robustness and general applicability of our framework.

We selected the strategies of lithium vacancy creation and anion substitution for two key reasons: first, they provide a straightforward approach to elucidating the concept of path entropy; second, these methods have been well measured and recorded through experimental studies. To this end, our methodology specifically highlights two experimentally validated strategies for entropy-driven design: (1) **engineering of the flexible lithium coordination shell** (LCS) through vacancy creation or cation doping, and

(2) **engineering the rigid anion framework** (such as $[\text{PS}_4]^{3-}$) through substitution or doping.

Importantly, the two approaches have been experimentally demonstrated to significantly enhance superionic conductivity, which arise many attentions. [*Nat. Energy*. **1**, 16030 (2016); *Angew. Chem. Int. Ed.* **58**, 8681-8686 (2019)] In contrast, many alternative strategies yield limited improvements [*Energy Environ. Mater.* **7**, e12729 (2024); *Energy Storage Mater.* **45**, 484-493 (2022)] compared to these two approaches. By focusing on these two well-established, high-impact strategies (substitution and vacancy engineering), we have constructed representative argyrodite systems (LPSCI-III and LSPSCI) that demonstrate the practical applicability of path entropy in real materials.

We also strongly align with your suggestion to extend validation through additional methodologies. However, for each strategy (e.g., lithium vacancy introduction, substitution, multicomponent integration ...), there are nearly infinite optimization pathways within systems like LPSCI, LGPS, and others. [*Joule* **6**, 543-587 (2022)] Therefore, demonstrating these two foundational entropy-driven aspects with solid experimental evidence provides a highly instructive, scalable foundation for future work. Nevertheless, to further validate the general applicability of path entropy, we have implemented a high-throughput screening approach targeting the broader sulfide compound space.

2. The authors defined escape entropy (S_e) as “the contribution from transitions from the original LCS to external states”, but failed to clarify the standard for defining the “original LCS” (i.e., how to define the boundaries of LCS). We suppose that relevant numerical standards can be added.

Re: Thank you for your careful examination and insightful comment regarding the fundamental role of LCS boundaries in our approach. The boundaries are determined through a Voronoi partition process, which we have provided details in **Supplementary Note 3**. In **Supplementary Note 3**, we have provided details of the method we used in determining the number of LCS and the partitioned space for each LCS. We have also provided numerical standards for a different cutoff range of LCS boundaries from 1.50 to 2.00 Å. The numerical values and standard errors are also provided in **Supplementary Tables 6 and 7**.

To make it clearer, we now have revised related parts in the revised manuscript with a short explanation of methods provided in **Supplementary Note 3**:

*“We then discretize each LCS into partitioned 3D subspaces representing discrete lithium states (LSs) using classical Voronoi partitioning (Fig. 2i, details in partitioned boundaries and error estimation are provided in **Supplementary Note 3**)”* (Lines 149-150)

Here’s the corresponding **Supplementary Note 3**:

“3. Supplementary Note 3

Periodic K-means and Voronoi partition.

To identify local coordination shells (LCSs), we implemented periodic K-means clustering algorithm using the scikit-learn package⁹. The number of LCSs, denoted as K (where K is less than or equal to the total number of lithium atoms in the unit cell), was determined by

minimizing the within-cluster sum of squares (WCSS). Following cluster identification, a Voronoi partition including application of periodic boundary conditions was performed in three dimensions for each LCS¹⁰. For Voronoi partitioning calculations, we defined a 3D box extending 1.75 Å from the maximum/minimum atomic positions of each LCS. Lithium atoms are considered to transition from an original LCS to a new LCS only when they move from the overlapping region of the original LCS to another LCS. Uncertainty analysis across this cutoff range (1.50–2.00 Å) was systematically evaluated and summarized in Supplementary Tables 6 and 7.”

3. Based on the last review, the gap of LCS among LPSCI-I, II, III, and LSPSCI (3/4/4/5 types) and whether it is related to the intrinsic structure of the materials are expected to be explained.

Re: Thank you for highlighting this important aspect of our approach. It’s related to the intrinsic structure of materials, especially the distribution of lithium ions. To ensure our method is well generalized to other materials, we use a periodic K-means algorithm to determine the number of LCS. We have provided the details of this method in **Supplementary Note 3**. To ensure clarity and accessibility, we have also made clarifications in the revised manuscript:

“Following a periodic K-means clustering method (Supplementary Note 3), which determines the optimal number of LCSs, lithium ions are assigned to LCSs characterized by distinct spatial and angular distributions.” (Lines 144-145)

4. In the flowchart that depicting the multistep high-throughput screening for identifying promising SSE candidates (Fig.5a), the way to determine the threshold setting of “MSD > 0.018 nm²” in Step 3 remains vague. Similarly, in Step 4, why “200 J/mol/K” was chosen as the critical value? Furthermore, is this value universal or specific for sulfide SSE systems?

Re: Thank you for your insightful comment regarding our screening criteria. We clarify the rationale for each as follows:

- (1) For the first MSD criterion, we selected this value (0.018 nm²) based on calibration with the argyrodite system Li₆PS₅Cl (1.92 × 10⁻² nm²; **Supplementary Table 9**). This slightly lower value ensures a margin of ~20 SSE candidates for baseline ionic diffusion performance. It is not a universal value but rather context-dependent: For initial screening with short-time MSD calculations involving tens of thousands of candidates, a higher threshold may be appropriate.
- (2) For the path entropy criterion, the 200 J/mol/K was chosen based on calibration with two systems: LPSCI-type (Li₆PS₅Cl; 273.52 ± 0.00 J/mol/K), and LGPS-type (Li₁₀Si(PS₆)₂; 167.51 ± 16.24 J/mol/K). This value is transferable across other systems when using the same machine learning potential (e.g., MACE). However, it is not universally fixed, it may be adjusted for looser or tighter selection criteria depending on high-throughput screening procedure.

5. It is necessary for authors to give an in-depth analysis of the fundamental relationship

among S_p , S_e and S_c . For example, several systems (e.g., LSPSCI) have both high configurational entropy and high path entropy, while others (e.g., LPSCI-III) show low configurational entropy but extremely high path entropy. Is this discrepancy related to the changes in lithium coordination environment?

Re: Thank you for your thoughtful question regarding the relationship among different types of entropies. The system-level disorder in SSE originates from diffusional disorder (quantified by S_p) and configurational disorder (quantified by S_c). Escape entropy S_e is part of S_p , representing entropic contribution of long-range diffusion.

In LSPSCI, substituting Si with P directly introduces configurational disorder into the host framework, resulting in high S_c . For LPSCI-III, lithium vacancies are localized within the LCSs rather than altering the host framework structure. Therefore, the configurational entropy change, relative to pristine LPSCI-II, is minimal.

Both systems exhibit high path entropy because these entropy-driven mechanisms, whether through substitution (LSPSCI) or vacancy introduction (LPSCI-III), effectively influence lithium-ion conduction within the LCSs. This conduction behavior is directly quantified by path entropy. Therefore, this discrepancy is related to the changes in LCS, whether the LCS has been directly engineered or not.

We have now added this detailed summary in the revised manuscript under section **Connection between configurational entropy and path entropy:**

“Entropy contributions related to ionic conduction in SSEs arise from both the flexible LCSs containing mobile lithium ions and the rigid host framework that forms the structural backbone. Disorder in these systems originates from two components: diffusional disorder associated with lithium-ion motion and configurational disorder in the host lattice. Consequently, total entropy is partitioned into path entropy S_p from the flexible LCSs and configurational entropy S_c from the rigid host framework. By combining these two components, the system-level disorder can be quantified rather than assessed solely through the disorder induced by the lithium-ion conduction or host framework configurations.

It is noteworthy that while systems with high S_p may exhibit high S_c , no direct causal relationship exists between these quantities. Instead, they represent complementary facets of solid-state ionic conductors. Joint analysis of S_p and S_c enables unambiguous identification of entropy gains in entropy-driven-designed systems. Compared to minor variations in the S_c by different entropy-driven strategies (e.g., vacancy introduction), the S_p , a metric quantifying diffusional disorder, aligns strongly with ionic diffusion performance. Its values range from 0.0 J/mol/K (LPSCI-I) to 415.72 ± 6.02 J/mol/K (LPSCI-II), 553.91 ± 54.42 J/mol/K (LSPSCI), and 2598.16 ± 112.58 J/mol/K (LPSCI-III) (Fig. 4g). These results confirm that the S_p provides a more direct and quantitatively robust metric for assessing ionic diffusion than the S_c . A more nuanced separation of escape entropy S_e from path entropy S_p , enables the clear identification of long-range diffusion in LSPSCI (91.70 ± 15.82 J/mol/K) and LPSCI-III (513.44 ± 26.86 J/mol/K).”

6. For the novel candidate material $\text{Li}_4\text{Cr}_2\text{C}_4\text{SO}_{16}$, only ionic conductivity at 300 K is

provided. To determine its activation energy and high-temperature stability, the temperature dependant EIS (e.g., Arrhenius plot) is suggested to supplement.

Re: Thank you for this constructive comment. As previously noted, the diffusion coefficients plot for $\text{Li}_4\text{Cr}_2\text{SO}_{16}$ was provided in the inset of **Fig. 5b**. Following your suggestion, we have now explicitly added the Arrhenius plot for this material in **Supplementary Fig. 44**, with a calculated activation energy of 0.18 eV. This supplementary figure is now explicitly referenced in the revised manuscript:

In the revised manuscript, the supplementary Arrhenius plot is referenced: “ $\text{Li}_4\text{Cr}_2\text{C}_4\text{SO}_{16}$ achieves ionic conductivity of 5.05 ± 0.23 mS/cm (Supplementary Fig. 44; activation energy 0.18 eV).” (Line 318)

Supplementary Figure 44. Arrhenius plot of ionic conductivities of $\text{Li}_4\text{Cr}_2\text{C}_4\text{SO}_{16}$. The corresponding activation energy is 0.18 eV.

7. please confirm whether the error value exceeds the average within the value “ $8.42 \pm 9.34 \times 10^{-2}$ mS/cm”. If it is wrong, please correct it. Otherwise, please explain the reason for the excessive error.

Re: Thank you for this careful examination and identifying this misleading expression. The accurate value is $8.42 \pm (9.34 \times 10^{-2})$ mS/cm (not $(8.42 \pm 9.34) \times 10^{-2}$ mS/cm). We have now corrected this in the revised manuscript:

“... and 8.42 ± 0.093 mS/cm (close to reported value⁹ of 9.4 mS/cm), respectively.” (Line 112)

“Notably, this conductivity rivals that of LPS-III (8.42 ± 0.093 mS/cm)” (Line 319)

We would like to thank Reviewer #2 again for the useful suggestions which are important for the improvement of our work.

Reviewer #3 (Remarks to the Author):

In this manuscript, the authors introduce SP as a novel descriptor to quantify the diffusional disorder of lithium ion pathways in SSEs, using a combination of the MSM, TPT and NNMD. The goal is to establish SP as a universal design principle for HTS of high-performance SSEs. The integration of information theory to analyze the complexity of ion diffusion pathways is a potentially transformative approach, offering a thermodynamic perspective on transport dynamics. Here are some comments:

Reply: We sincerely thank Reviewer #3 for his/her detailed review and insightful comments on our approach, which is potentially transformative for analyzing transport dynamics. Below, we have carefully addressed all questions Reviewer #3 has raised, with original manuscript sentences underlined and **revised text or new results highlighted in red**.

1. The electrochemical stability is the primary hurdle for practical SSEs. The work solely focuses on transport dynamics. Any proposed material must be stable under synthesis and operating conditions. The manuscript completely lacks analysis of stability of the predicted phases.

Re: Thank you for pointing out the very fundamental property of SSE should be considered during the high-throughput screening. We fully agree that electrochemical stability represents the primary hurdle for practical SSEs. Nevertheless, we employ the energy above hull (E_{hull}) which is well accepted to be the essential metric for assessing electrochemical stability, hence directly evaluating the stability of all screened SSEs. We have listed the E_{hull} for all candidates meeting the criterion $E_{\text{hull}} < 0.085$ eV/atom (comparable to $\text{Li}_6\text{PS}_5\text{Cl}$ system of 0.083 eV/atom) (**Supplementary Table 9**). Since this stability criterion was implemented in Step 2 of our high-throughput screening, all predicted candidates satisfy $E_{\text{hull}} < 0.085$ eV/atom, confirming their computational stability.

To make it clearer, in the revised manuscript, the procedure we've implemented in our manuscript has been explicitly described:

“Subsequent filtering criteria included: (i) a band gap greater than 2.0 eV ($E_{\text{band}} \geq 2.0$ eV) and (ii) an energy above hull (E_{hull}) ≤ 85 meV per atom (comparable to LPSCL-II). This reduced the candidate pool to 509 materials. While these candidates satisfied the initial screening criteria for structural and electronic stability, ionic mobility remains a critical factor for SSE performance.”

2. The manuscript identifies $\text{Li}_4\text{Cr}_2\text{C}_4\text{SO}_{16}$ as a high-performance candidate, yet its calculated diffusion disorder is reported as low, similar to the poorly conducting reference phase. This contradiction must be resolved through deep microstructural and mechanistic analysis.

Re: Thank you for your careful review and this insightful comment. Following your suggestion, we have now resolved it with additional description and mechanistic analysis of this new compound.

The calculated path entropy of $\text{Li}_4\text{Cr}_2\text{C}_4\text{SO}_{16}$ is 263.98 ± 10.42 J/mol/K, comparable to the $\text{Li}_6\text{PS}_5\text{Cl}$ (LPSCI-II) at 273.52 ± 0.00 J/mol/K. Crucially, the ionic conductivity of the novel $\text{Li}_4\text{Cr}_2\text{C}_4\text{SO}_{16}$ (5.05 ± 0.23 mS/cm) is significantly higher than that of LPSCI-II ($1.53 \times 10^{-3} \pm 1.40 \times 10^{-4}$ mS/cm).

This increased ionic conductivity arises from a greater number of lithium conduction states with a wider distribution relative to LPSCI-II. Specifically, $\text{Li}_4\text{Cr}_2\text{C}_4\text{SO}_{16}$ exhibits 5 connected LCSs, enabling efficient transfer between states (**Supplementary Fig. 45**). As shown in **Supplementary Fig. 46-50**, comparing with the kinetics of lithium conduction in LPSCI-II or LPSCI-III, the mean first passage time (MFPT) profile of $\text{Li}_4\text{Cr}_2\text{C}_4\text{SO}_{16}$ lies between LPSCI-II and LPSCI-III. This intermediate behavior occurs because the connected LCSs permit unforbidden out-LCS conduction (unlike LPSCI-II; **Supplementary Fig. 14-17**) but without the intensity of conduction in LPSCI-III (**Supplementary Fig. 18-21**). Consequently, the ionic conduction in this material is less constrained than in LPSCI-II.

These results are now fully incorporated into **Supplementary Fig. 45-50**. A detailed microstructural and mechanistic analysis has also been added to the revised manuscript: *“Notably, this conductivity rivals that of LPSCI-III (8.42 ± 0.093 mS/cm), where long-range ionic diffusion is activated via lithium-ion vacancies and site disorder. $\text{Li}_4\text{Cr}_2\text{C}_4\text{SO}_{16}$ exhibits 32 lithium sites with sparse distribution compared to LPSCI-II, resulting in 5 connected LCSs (**Supplementary Fig. 45**). This LCS configuration enables enhanced lithium-ion transport kinetics, which fall between the constrained kinetics of LPSCI-II (**Supplementary Fig. 14-17**) and the fully activated diffusion in LPSCI-III (**Supplementary Fig. 18-21**). The partially unlocked out-LCS conduction facilitates higher ionic conductivity (**Supplementary Fig. 44**) despite comparable path entropy to LPSCI-II. The nearly-zero S_e (1.8 J./mol/K; **Supplementary Table 9**) in $\text{Li}_4\text{Cr}_2\text{C}_4\text{SO}_{16}$ confirms predominantly short-range diffusion mechanisms, analogous to pristine LPSCI-II. This characteristic suggests that employing entropy-driven strategies (e.g., vacancy engineering or site disorder) shown in step 6 of **Fig. 5a** could further significantly enhance its superionic performance.”* (Lines 319-332)

Supplementary Figure 45. Distribution of LCSs and path entropies in $\text{Li}_4\text{Cr}_2\text{C}_4\text{SO}_{16}$. **a**, distribution of LCSs in $\text{Li}_4\text{Cr}_2\text{C}_4\text{SO}_{16}$. **b**, bar plot of LCS-based path entropy and escape entropy.

Supplementary Figure 46. MFPT profile of lithium hopping in $\text{Li}_4\text{Cr}_2\text{C}_4\text{SO}_{16}$ at 300 K of LCS-1. The initial lithium states in LCS-1 are denoted with a cyan rectangle. Blocks colored blue represent no transition happens or have MFPT value greater than 10000 ps.

Supplementary Figure 47. MFPT profile of lithium hopping in $\text{Li}_4\text{Cr}_2\text{C}_4\text{SO}_{16}$ at 300 K of LCS-2. The initial lithium states in LCS-2 are denoted with a cyan rectangle. Blocks colored blue represent no transition happens or have MFPT value greater than 10000 ps.

Supplementary Figure 48. MFPT profile of lithium hopping in Li₄Cr₂C₄SO₁₆ at 300 K of LCS-3. The initial lithium states in LCS-3 are denoted with a cyan rectangle. Blocks colored blue represent no transition happens or have MFPT value greater than 10000 ps.

Supplementary Figure 49. MFPT profile of lithium hopping in Li₄Cr₂C₄SO₁₆ at 300 K of LCS-4. The initial lithium states in LCS-4 are denoted with a cyan rectangle. Blocks colored blue represent no transition happens or have MFPT value greater than 10000 ps.

Supplementary Figure 50. MFPT profile of lithium hopping in $\text{Li}_4\text{Cr}_2\text{C}_4\text{SO}_{16}$ at 300 K of LCS-5. The initial lithium states in LCS-5 are denoted with a cyan rectangle. Blocks colored blue represent no transition happens or have MFPT value greater than 10000 ps.

3. The accuracy of MSM is sensitive to input parameters. The manuscript must prove the statistical robustness of its underlying simulations.

Re: We fully agree with your point that robustness and reproducibility form the foundation of a new approach. To ensure the robustness of our methodology, particularly the MSM construction and associated Voronoi partition analysis, we conducted three independent simulations, each with a minimum trajectory length of 1.0 ns, as input for MSM modeling. The final path entropy and escape entropy values are derived from aggregating results across all three simulations.

(1) The statistical validation of this robust approach is detailed in the Method section **Calculation of lithium flux and path entropy:**

“... For four types of SSE, three independent simulations were performed to compute probability densities and associated path entropies. The mean path entropies fall within a 99.5% confidence interval.” (Lines 484-486)

(2) For parameters affecting MSM modeling (e.g., Voronoi partition boundaries), we report standard errors by counting the numerical values of partition boundaries (**Supplementary Note 3**):

“Periodic K-means and Voronoi partition.”

*To identify local coordination shells (LCSs), we implemented periodic K-means clustering algorithm using the scikit-learn package⁹. The number of LCSs, denoted as K (where K is less than or equal to the total number of lithium atoms in the unit cell), was determined by minimizing the within-cluster sum of squares (WCSS). Following cluster identification, a Voronoi partition including application of periodic boundary conditions was performed in three dimensions for each LCS¹⁰. For Voronoi partitioning calculations, we defined a 3D box extending 1.75 Å from the maximum/minimum atomic positions of each LCS. Lithium atoms are considered to transition from an original LCS to a new LCS only when they move from the overlapping region of the original LCS to another LCS. Uncertainty analysis across this cutoff range (1.50–2.00 Å) was systematically evaluated and summarized in **Supplementary Tables 6 and 7.**”*

These measures confirm the reproducibility and reliability of our analysis framework.

4. Provide a more in-depth microphysical explanation for the complementarity between structural configurational entropy and pathway entropy.

Re: Thank you for your insightful suggestion to provide a detailed explanation of path entropy and configurational entropy. As we illustrated in **Fig. 1**, SSEs can be decomposed into two distinct components: (1) A flexible local coordination shell (LCS) containing mobile lithium ions, and (2) A rigid host framework. This decomposition allows us to separate total disorder into two physically meaningful contributions:

- (1) Configurational entropy S_c quantifies disorder arising from the rigid host framework.
- (2) Path Entropy S_p quantifies disorder associated with the flexible LCS, representing the diffusional disorder of lithium ions.

By explicitly distinguishing these components, we can now assess total disorder in ionic systems, not merely lithium conduction effects or host framework distortions.

We have now incorporated this in-depth explanation in the revised manuscript under section **Connection between configurational entropy and path entropy**:

“Entropy contributions related to ionic conduction in SSEs arise from both the flexible LCSs containing mobile lithium ions and the rigid host framework that forms the structural backbone. Disorder in these systems originates from two components: diffusional disorder associated with lithium-ion motion and configurational disorder in the host lattice. Consequently, total entropy is partitioned into path entropy S_p from the flexible LCSs and configurational entropy S_c from the rigid host framework. By combining these two components, the system-level disorder can be quantified rather than assessed solely

through the disorder induced by the lithium-ion conduction or host framework configurations.

It is noteworthy that while systems with high S_p may exhibit high S_c , no direct causal relationship exists between these quantities. Instead, they represent complementary facets of solid-state ionic conductors. Joint analysis of S_p and S_c enables unambiguous identification of entropy gains in entropy-driven-designed systems. Compared to minor variations in the S_c by different entropy-driven strategies (e.g., vacancy introduction), the S_p , a metric quantifying diffusional disorder, aligns strongly with ionic diffusion performance. Its values range from 0.0 J/mol/K (LPSCI-I) to 415.72 ± 6.02 J/mol/K (LPSCI-II), 553.91 ± 54.42 J/mol/K (LSPSCI), and 2598.16 ± 112.58 J/mol/K (LPSCI-III) (Fig. 4g). These results confirm that the S_p provides a more direct and quantitatively robust metric for assessing ionic diffusion than the S_c . A more nuanced separation of escape entropy S_e from path entropy S_p , enables the clear identification of long-range diffusion in LSPSCI (91.70 ± 15.82 J/mol/K) and LPSCI-III (513.44 ± 26.86 J/mol/K).”

We would like to thank Reviewer #3 again for the useful suggestions which are important for the improvement of our work.

List of Changes

1. Lines 144-149: **Supplementary Note 3** now explicitly describes the methods for the analysis.
2. Line 151: The number of lattice sites (LSs) has been clarified to avoid ambiguity.
3. Lines 233-236: A detailed explanation of rotation behaviours in LSPSCI has been added.
4. Section “**Connection between configurational entropy and path entropy**”: Additional analysis (lines 256-264 and 274-277) provides an in-depth explanation of the relationship between different entropies.
5. Line 318: The Arrhenius plot (**Supplementary Fig. 44**) is now added and includes the calculated activation energy. Supplementary data has been updated accordingly.
6. Lines 321-328: The high ionic conductivity of $\text{Li}_4\text{Cr}_2\text{C}_4\text{SO}_{16}$ is now fully explained with new figures (**Supplementary Fig. 45-50**: LCS and MFPT profiles). Supplementary data has been updated.
7. Line 458: Dynamic processes for LPSCI-II, LPSCI-III, and LSPSCI (**Supplementary Fig. 51-53**) are now referenced, with related figures provided in Supplementary data.
8. Lines 482-484: The description of standard error calculation has been clarified.
9. Supplementary data (lines 330-332): Details of Voronoi partitioning and diffusion between LCSs have been clarified.

Manuscript ID: NCOMMS-25-93405A

Title: "Path Entropy-driven Design of Solid-State Electrolytes"

Author(s): Qiye Guan, Kaiyang Wang, Jingjie Yeo, Yongqing Cai

February 18, 2026

Response to Reviewers

Reviewer #1 (Remarks to the Author):

Dear Authors,

thank you for thoroughly addressing all comments and making all necessary changes to the manuscript and supporting information. I suggest accepting the manuscript for publication.

Reply: We sincerely appreciate your recommendation for acceptance of our revised manuscript.

Reviewer #2 (Remarks to the Author):

The authors have addressed the comments well. I recommend that this manuscript could be published in Nature Communications.

Reply: We sincerely appreciate your recommendation for acceptance of our revised manuscript.

Reviewer #3 (Remarks on code availability):

The authors have addressed several of my previous concerns. However, before the manuscript can be recommended for publication, there are two critical technical points regarding the simulation settings and data reliability that must be clarified:

Reply: We sincerely appreciate **Reviewer #3** for his/her detailed review and thorough examination of our simulations. Below, we have carefully addressed all questions **Reviewer #3** has raised, with original manuscript sentences underlined and **revised text or new results highlighted in red**.

1. In the SI, the authors stated that the energy cutoff for the projector augmented wave method was set to 450 eV. However, for systems containing Li, the standard PAW potentials typically recommend a maximum cutoff of approximately 499.034 eV for high-precision calculations. Could the authors justify the use of 450 eV, which is lower than the recommended value for Li? Please provide evidence to demonstrate that this choice does not compromise the accuracy.

Reply: Thank you for this careful observation. We fully agree that simulations must be computed with maximal accuracy when possible. We would like to clarify that there exist two types of pseudopotentials generated for Li depending on the selection of valence electrons ($2s^1$ or $1s^2 2s^1$). We adopted the standard Li PAW potential with $2s^1$ (recommended energy max (ENMAX) = 140 eV) in our calculations. Our energy cutoff of 450 eV is approximately $3.2 \times \text{ENMAX}$ for the standard Li potential (ENMAX = 140 eV) and exceeds the ENMAX values of all elements in the system, ensuring well-converged results for structural stability. Regarding the slightly higher cutoff value of 499.034 eV mentioned by the Reviewer, it corresponds to the Li_sv ($1s^2 2s^1$) potential, which treats the 1s electrons as valence states.

This calculation setting (450 eV cutoff with standard Li potential) was selected for identifying stable LPSCI-III structures via cluster-expansion (CE), as detailed in the Methods section. For CE model preparation, we screened 340 structures generated through enumeration. Given our focus on structural properties rather than formation energies or electronic structures, the valence 2s electron is most relevant and standard Li potential is computationally efficient for this large dataset. This approach avoids unnecessary cost while maintaining sufficient accuracy for relative energy comparisons.

Nevertheless, to justify this choice, as inspired by the Reviewer, we had also tested 10 randomly selected structures simulated and compared with both standard Li (energy cutoff = 450 eV) and Li_sv (energy cutoff = 500 eV) pseudopotentials. As shown in **Figure R1**, both settings consistently distinguish stable structures, with deviations mostly limited to absolute energy values. Thus, the standard Li PAW potential (ENMAX= 140 eV) at 450 eV energy cutoff provides a cost-effective and reliable basis for our CE model initialization.

Fig. R1 Comparison of energies based on two different versions of Li pseudopotentials under different cutoffs for identifying stable LPSCI-III structures.

2. On lines 302-303, the author stated “Candidates with measurable lithium-ion mobility, defined as MSDs > 0.018 nm² over 6.0 ps of NNMD simulation”. A 6.0 ps window is relatively short for extracting diffusion properties in complex solid-state electrolytes. More importantly, the authors must ensure that the system has reached both thermodynamic and kinetic equilibrium before the commencement of the MSD data collection.

Reply: We sincerely appreciate your insightful comment regarding the MSD calculation. As suggested by the Reviewer, we conduct more examination to evaluate the issue of time window to derive the MSD. We indeed found that a 6.0 ps window is relatively insufficient for high-throughput screening and therefore have implemented the following improvements:

- (1) Extended simulation window: We conducted a 12.0 ps NNMD simulation for all the candidates with the first 2.0 ps discarded to ensure full thermal relaxation at room temperature prior to analysis.
- (2) Revised screening criterion: To prevent omission of potential superionic candidates, we lowered the MSD threshold from 0.018 nm² to 0.010 nm² in the third screening round. This adjustment accounts for systematic overestimation of MSD due to the previously relatively short simulation window and incomplete system relaxation.

As a result, the candidate list after MSD calculation was expanded from 20 to 27 compounds (See **Supplementary Table 9**), in which 17 compounds were retained from the previous list and 10 compounds are newly added highlighted in red. Subsequent path entropies (S_p) calculations of these 10 new candidates identified three experimentally confirmed high-performance superionic conductors with $S_p > 200$ J/mol/K: (1) Argyrodite-type: $\text{Li}_6\text{PS}_5\text{Br}$ and Li_7PS_6 [*ACS Appl. Mater. Interfaces*, **11**, 6015–6021 (2019); *Chem. Eur. J.* **16**, 5138–5147 (2010)] and (2) Thioborate-type [*J. Mater. Res.* **37**, 3269–3282 (2022)] $\text{Li}_5\text{B}_7\text{S}_{13}$ [*Solid State Ion.* **78**, 305–313 (1995)]. Additionally, we also identified four other experimentally or theoretically validated candidates with lower S_p values: Li_4TiS_4 [*J. Am. Chem. Soc.* **139**, 8796–8799 (2017)], Li_3CuS_2 [*ACS Appl. Energy Mater.* **4**, 20–24 (2020)], Li_4GeS_4 [*Solid State Ion.* **154-155**, 789–794 (2002)], and Li_8TiS_6 [*Npj Comput. Mater.* **11**, 67 (2025)]. These results are now fully incorporated into **Fig. 5** and **Supplementary Table 9**.

We have revised the manuscript accordingly regarding the updated results:

“Candidates with measurable lithium-ion mobility, defined as MSDs > 0.010 nm² over 12.0 ps of NNMD simulation, were retained. This procedure narrowed the list to 27 structurally stable compounds with detectable ionic motion (Supplementary Table 9).” (Lines 303-305)

“..., with the $S_p > 200.0$ J/mol/K, including experimentally verified SSEs such as argyrodite-type compounds^{14,29,30} (LPSCI-II ($\text{Li}_6\text{PS}_5\text{Cl}$), Si-substituted LSPSCI^{13,14} ($\text{Li}_{20}\text{Si}_3\text{P}_3\text{S}_{23}\text{Cl}_1$), $\text{Li}_6\text{PS}_5\text{Br}$, and Li_7PS_6 ⁴⁰), thioborate-type⁴¹ $\text{Li}_5\text{B}_7\text{S}_{13}$ ⁴², $\text{Li}_7\text{P}_3\text{S}_{11}$ ⁴³, rock-salt sulfide Li_3NbS_4 ⁴⁴, and LGPS-type structures⁴⁵ ($\text{Li}_{10}\text{Ge}(\text{PS}_6)_2$, $\text{Li}_{10}\text{Sn}(\text{PS}_6)_2$).” (Lines 309-313)

The high-throughput screening procedure described in **Supplementary Note 6.2** has also been revised:

“..., NNMD simulations were conducted over a total time of 12.0 ps using a 2×2×2 supercell. To ensure the system reaches full equilibrium at room temperature, the first 2.0 ps of simulated trajectories were discarded before MSD analysis. To generate path entropies for the final production (4th screening) run encompassing all 27 candidate materials, NNMD simulations were performed for at least 800.0 ps using larger supercells (typically a 3×3×3 supercell, varying depending on cell size).” (Lines 389-394)

Fig. 5. High-throughput screening of potential solid-state electrolytes based on path entropy. a, Flowchart depicting the multistep high-throughput screening procedure for identifying promising SSE candidates. b, Summary of the path entropy and the escape entropy screening results (top 20 out of 27 candidates) from the step 4 of the workflow. Candidates with favorable lithium-ion diffusion characteristics are highlighted with cyan rectangles. The inset shows diffusion coefficient D as a function of the reciprocal temperature for $\text{Li}_4\text{Cr}_2\text{C}_4\text{SO}_{16}$.

Supplementary Table 9. Summary of 27 screened superionic candidates.

No.	Identifier	Space Group	Band gap (eV)	Energy above hull per atom (eV)	Chemical formula	MSD at 300K (nm ²)	Path entropy (J/mol/K)	Escape entropy (J/mol/K)
1	mp-1040451	6	2.39	0.035	$\text{Li}_{20}\text{Si}_3\text{P}_3\text{S}_{23}\text{Cl}$	0.0118	1016.57 ± 27.10	254.69 ± 10.30
2	mp-532413	9	3.59	0.000	$\text{Li}_5\text{B}_7\text{S}_{13}$	0.0165	629.35 ± 63.38	102.84 ± 22.50
3	agm003282504	105	2.09	0.021	$\text{Li}_{10}\text{Ge}(\text{PS}_6)_2$	0.0156	381.37 ± 51.93	76.60 ± 19.55
4	mp-641703	2	2.49	0.020	$\text{Li}_7\text{P}_3\text{S}_{11}$	0.0178	314.62 ± 18.74	50.73 ± 1.43
5	mp-696123	105	2.27	0.035	$\text{Li}_{10}\text{Sn}(\text{PS}_6)_2$	0.0156	300.41 ± 17.32	46.25 ± 1.64
6	mp-769048	62	2.15	0.078	Li_3NbS_4	0.0117	298.29 ± 15.32	46.02 ± 6.44
7	mp-985592	216	2.30	0.083	$\text{Li}_6\text{PS}_5\text{Cl}$	0.0243	273.52 ± 0.00	0.00 ± 0.00
8	mp-771912	70	2.30	0.070	$\text{Li}_4\text{Cr}_2\text{C}_4\text{SO}_{16}$	0.0141	263.98 ± 10.42	1.80 ± 0.00
9	mp-985591	216	2.14	0.051	$\text{Li}_6\text{PS}_5\text{Br}$	0.0162	253.08 ± 0.00	0.00 ± 0.00
10	mp-1211324	33	2.08	0.033	Li_7PS_6	0.0119	224.56 ± 42.84	20.67 ± 5.66
11	mp-768991	70	2.05	0.076	$\text{Li}_4\text{Bi}_2\text{C}_4\text{SO}_{16}$	0.0128	189.82 ± 25.98	1.51 ± 0.00
12	agm003282544	105	2.41	-0.008	$\text{Li}_{10}\text{Si}(\text{PS}_6)_2$	0.0186	167.51 ± 16.24	17.94 ± 7.20
13	mp-766600	218	2.42	0.069	Li_4TiS_4	0.0196	166.70 ± 18.89	3.92 ± 2.97

14	agm00323 2549	141	2.02	0.077	Li ₃ CuS ₂	0.0165	95.80 ± 12.33	1.76 ± 1.64
15	mp- 1222582	33	2.51	0.026	Li ₄ GeS ₄	0.0220	65.66 ± 0.89	0.70 ± 0.14
16	mp- 766575	33	2.70	0.063	Li ₈ TiS ₆	0.0184	50.37 ± 24.03	0.93 ± 1.82
17	agm00323 0254	7	4.74	0.041	LiH ₃ (SeO ₃) ₂	0.0251	22.39 ± 0.06	0.00 ± 0.00
18	mp- 849740	1	2.08	0.028	LiFe(SO ₄) ₂	0.0135	17.78 ± 0.00	1.41 ± 0.00
19	agm00323 5290	62	4.39	0.064	CsLiSO ₄	0.0122	16.33 ± 0.00	0.00 ± 0.00
20	agm00221 9141	82	2.01	0.030	LiCdPSe ₄	0.0246	4.14 ± 0.00	0.00 ± 0.00
21	mp- 769554	1	2.43	0.009	LiMgCr ₃ (SO ₄) ₆	0.0241	0.00 ± 0.00	0.00 ± 0.00
22	mp- 754902	4	4.06	0.068	LiSbCSO ₇	0.0220	0.00 ± 0.00	0.00 ± 0.00
23	mp- 695469	1	2.25	0.028	Li ₃ Cr ₁₃ Ni ₃ (SO ₄) ₂₄	0.0202	0.00 ± 0.00	0.00 ± 0.00
24	mp- 769549	1	2.32	0.011	LiZnCr ₃ (SO ₄) ₆	0.0193	0.00 ± 0.00	0.00 ± 0.00
25	mp- 770759	4	3.75	0.070	LiBiCSO ₇	0.0174	0.00 ± 0.00	0.00 ± 0.00
26	mp- 766088	1	2.33	0.009	LiCr ₃ Ni(SO ₄) ₆	0.0179	0.00 ± 0.00	0.00 ± 0.00
27	agm00234 2398	164	2.04	0.072	LiS ₂ Sm	0.0124	0.00 ± 0.00	0.00 ± 0.00

We thank **Reviewer #3** again for these two suggestions, which are important for the improvement of our work.

List of changes

1. Lines 283–287 and 312: Results of high-throughput screening have been updated in **Fig. 5** and related analysis; **Supplementary Table 9** (line 280) has been revised accordingly.
2. Lines 303–305: Details of MSD calculation methodology have been updated.
3. Supplementary data (lines 389–394): Descriptions of NNMD simulations and associated MSD analysis have been revised.

Summary of Comments on 1_reviewer_attachment_1_1766403402_convrt.pdf

Page: 5

H Author: redacted Subject: Comment on Text Date: 18/12/2025 10:13:11
This is a computationally identified material. What is the main limitation for its synthesis and experimental validation?

H Author: redacted Subject: Comment on Text Date: 18/12/2025 10:20:46
Is this a substitution of addition of Si on unoccupied P site?

Page: 7

H Author: redacted Subject: Comment on Text Date: 18/12/2025 10:51:07
Are these total LSs or per LCS?

Page: 8

H Author: redacted Subject: Comment on Text Date: 18/12/2025 10:50:08
How many dynamic processes were identified for the other compositions? Is this information included somewhere else?

Page: 11

H Author: redacted Subject: Highlight Date: 18/12/2025 11:13:50

H Author: redacted Subject: Comment on Text Date: 18/12/2025 11:38:20
Please elaborate on the distinct rotation behavior, if a higher barrier was calculated.
Other factors previously calculated like the inter-LCS diffusion seem to be more relevant for enhancing ion conduction in LSPSCI than tetrahedra rotation?

Page: 14

H Author: redacted Subject: Comment on Text Date: 18/12/2025 11:45:34
Why do S_p and S_e for LSPSCI here differ from previous calculations shown in Fig.4?